# Mating harassment may boost the effectiveness of the sterile insect technique for *Aedes* mosquitoes

Dongjing Zhang [1,14], Hamidou Maiga [2,3,14], Yongjun Li [4,5,14], Mame Thierno Bakhoum [3,6,14], Gang Wang [1,14], Yan Sun[1], David Damiens[7], Wadaka Mamai[2], Nanwintoum Séverin Bimbilé Somda[2,8], Thomas Wallner[2], Odet Bueno-Masso[2], Claudia Martina[2], Simran Singh Kotla[2], Hanano Yamada[2], Deng Lu[9], Cheong Huat Tan [9], Jiatian Guo[1], Qingdeng Feng[1], Junyan Zhang[1], Xufei Zhao[1], Dilinuer Paerhande[1], Wenjie Pan[10], Yu Wu[1], Xiaoying Zheng[1], Zhongdao Wu[1], Zhiyong Xi [5,11], Marc J. B. Vreysen[2] & Jérémy Bouyer [2,12,13,14] ✉

The sterile insect technique is based on the overflooding of a target population with released sterile males inducing sterility in the wild female population. It has proven to be effective against several insect pest species of agricultural and veterinary importance and is under development for *Aedes* mosquitoes. Here, we show that the release of sterile males at high sterile male to wild female ratios may also impact the target female population through mating harassment. Under laboratory conditions, male to female ratios above 50 to 1 reduce the longevity of female *Aedes* mosquitoes by reducing their feeding success. Under controlled conditions, blood uptake of females from an artificial host or from a mouse and biting rates on humans are also reduced. Finally, in a field trial conducted in a 1.17 ha area in China, the female biting rate is reduced by 80%, concurrent to a reduction of female mosquito density of 40% due to the swarming of males around humans attempting to mate with the female mosquitoes. This suggests that the sterile insect technique does not only suppress mosquito vector populations through the induction of sterility, but may also reduce disease transmission due to increased female mortality and lower host contact.

The SIT is based on the sequential release of sterile male insects over the target area where they will mate with the wild female insects[1], resulting in the induction of sterility in the wild female population proportionally to the ratio of sterile to wild insects. This impairs the reproduction rate of the female population and as a result, fewer insects will be available in subsequent generations, reducing the density of the target population over time. The SIT has been successfully used to manage populations of various insect pests of agricultural, animal, or human health importance[2], and more recently, there has been a renewed interest to develop and implement the SIT against mosquitoes[3]. *Aedes* mosquitoes are major vectors of viruses such as dengue, chikungunya, Zika and yellow fever that are severely impacting human health. Traditional vector control strategies such as the use of broad-spectrum insecticides have serious environmental drawbacks and sanitation through reduction or removal of mosquito breeding sites requires the collaboration of the resident human population and has limited impact[4,5]. In 2023, 42 SIT pilot projects were being implemented worldwide against mosquitoes[6]. Released males are attracted by hosts, including humans[7], and can swarm around them in the search of mates, a behaviour that is exploited to monitor their density

through the Human Landing Catch method[8]. Alternatively, they can be trapped using $CO_2$-baited adult traps[9]. Continuous, inundative releases of sterile males, like those required for SIT, can lead to high sterile-to-wild male and male-to-female ratios, sometimes over 100 to 1, particularly when the target population is suppressed. Could such high sex ratios have some influence on the fitness of females?

Mating is an essential component of adult life for all species with sexual reproduction. In most insects, a single or a moderate number of matings are sufficient for females to maximize their reproductive success[10–12]. Therefore, females generally prefer a lower mating rate than males[13] and are often resistant or reluctant to re-mate[14]. This apparent divergence leads males from a wide range of animal species to compel females to mate by coercion or harassment[15]. As a consequence, a ratio of 10 sterile male *Aedes aegypti* to 1 female resulted in increased mortality of the females but did not impact the fitness of the surviving ones[12]. It was also suggested that in *Anopheles gambiae*, exposure to males, rather than consequences of mating, may reduce female longevity[16]. Mating harassment is a form of sexual conflict where repeated attempts to copulate by the male can be costly for the female[15]. These costs can be direct (effects on harassed females) or indirect (effects on descendants of harassed females)[12]. Harassment behaviours are even more frequent when individuals are confined to closed environments, like a rearing cage in the laboratory. Under mass-rearing conditions for example, a reduced 1:3 male to female ratio is recommended to reduce mating harassment and maximize production in both *Ae. albopictus*[17,18] and *Ae. aegypti*[19]. The same applies to other insects like tsetse flies where a 1:4 male to female ratio increases female fecundity in *Glossina fuscipes fuscipes* and *G. pallidipes*[20]. However, the effects of large sex ratios such as those observed during an SIT programme are largely unknown.

Here, we show that mating harassment by sterile male mosquitoes reduces the survival and feeding success of *Ae. albopictus* and *Ae. aegypti* females under laboratory, semi-field and field conditions.

## Results

### Survival of mosquitoes caged at different sex ratios

All experiments aiming to measure survival were done at a constant density of mosquitoes per cage, only varying the sex ratio, in order to control density-dependent mortality. We first observed the effect of high fertile male-to-female ratios in *Ae. aegypti* and *Ae. albopictus* in confined laboratory cages. In both species, increased male-to-female ratios were associated with higher mortality of the females and also of male *Ae. albopictus* (Supplementary Figs. 1–3). Even with a male-to-female ratio of 3:7, which is only slightly higher than the control at 1:3, mortality of female *Ae. aegypti* significantly increased (Supplementary Fig. 2, Supplementary Table 1, $P = 0.021$). Female mortality reached 14.5% (SD = 3.9%) after 8 days under a male-to-female ratio of 99:1 as compared with 2.8% (SD = 1.2%) in the control group (male-to-female ratio of 1:3). The impact of harassment on the survival of female *Ae. albopictus* was even more pronounced than in *Ae. aegypti*. A male-to-female ratio of 50:1 was enough to increase mortality of females significantly after 8 days (Supplementary Fig. 1, Supplementary Table 1, $P < 10^{-4}$), i.e., 38.9% (SD = 1.9%), similarly to under a male to female ratio of 100:1, whereas in the control group mortality remained at 1.5%. At the beginning of the experiment, we monitored some of the non-irradiated groups up to 13 days and mortality of females reached 43.3% (SD = 4.7%) in the 99:1 batch in comparison to 4.1% (SD = 1.7%) in the 1:3 control group (Supplementary Fig. 3, $P < 10^{-4}$).

Fertile male *Ae. aegypti* did not experience increased mortality with increased sex ratios (Supplementary Figs. 1, 2, Supplementary Table 1, $P > 0.05$). On the contrary, the mortality of male *Ae. albopictus* also increased with a male-to-female ratio of 50:1 after 8 days (Supplementary Fig. 1, Supplementary Table 1, $P < 10^{-4}$), i.e. 19.0% (SD = 4.2%), similarly to the batch with a male-to-female ratio of 100:1,

whereas in the control group mortality remained at 2.9%. This may be related to male *Ae. albopictus* being more aggressive, but this will warrant further research.

A practical application is that the increase in female mortality could be used as an additional process to separate the sterile males from the females by keeping them for some days in the insectary following mechanical separation that results in 1% or more female contamination of the sterile male batches[21]. We thus repeated the same experiments with irradiated mosquitoes to assess whether similar results would be obtained. In general, irradiation exacerbated the negative impact of mating harassment (Fig. 1). In irradiated mosquitoes, the cumulative mortality rate of female *Ae. aegypti* increased with sex ratio and was 26.7% (SD = 14.0%) after 8 days for a male-to-female ratio of 99:1 as compared with a mortality rate of 3.9% (SD = 2.4%) in the control group (Fig. 1 and Supplementary Table 2, $P < 10^{-4}$). Male *Ae. aegypti* mortality after 8 days did not increase with a male-to-female ratio of 99:1 (Supplementary Fig. Table 2, $P = 0.115$) and was even lower than in the control group for a ratio of 49:1. The cumulative mortality of *Ae. albopictus* (Reunion strain) females was even higher as compared with *Ae. aegypti* and reached 40.0% (SD = 8.8%) after 8 days with a male-to-female ratio of 100:1 as compared with 3.8% in the control group (Fig. 1 and Supplementary Table 2, $P < 10^{-4}$). Again, the cumulative mortality of males significantly increased with increasing sex ratio in this species, reaching 24.8% (SD = 0.61%) and 25.3% (SD = 4.1%) after 8 days with male to female ratios of 50:1 and 100:1, respectively, as compared with 2.9% in the control group. Comparable results were obtained in a similar trial with another strain of *Ae. albopictus* (Rimini), except that the mortality of females reached 90% (SD = 6.1%) after 8 days with a female-to-male ratio of 99:1 as compared with 17.3% (SD = 4.1%) in the control group (Supplementary Fig. 4, $P < 10^{-4}$). Caging of sterile males and females under laboratory conditions at a sex ratio of 100:1 thus decreased the female contamination of the sterile male batches to -0.6% and 0.7% due mortality for female *Ae. aegypti* and *Ae. albopictus*, respectively, within the first eight days. When a predetermined threshold is agreed with the public health authorities, e.g., 1%[21], this might be an effective way of eliminating females instead of removing residual females manually or discarding the full batch of sterile males. Nevertheless, this would probably be cost-prohibitive in an operational programme (see Supplementary Discuss).

### What causes mortality in high male-to-female ratios?

To better understand the mechanisms leading to increased mortality, we filmed the sexual interactions of the mosquitoes at a high resolution (1080 P). Females were harassed when sex ratios were biased towards males (see Supplementary Movie 1). At the highest male-to-female ratio of 99:1, females were completely prevented from feeding and were lying immobile at the bottom of the cage to escape further mating attempts from males who were aggregated around the females by groups of three to five individuals. Any attempt of females to escape attracted more males, probably induced by their wing beat. To verify this hypothesis, some females were glued on their back to a pin (see Supplementary Movie 2), and those females accepted two or three mates, but refused to re-mate thereafter. However, each time they were trying to escape and fly off, new males were attracted and were aggregating around them.

From these mosquito recordings, it was clear that feeding inhibition was the main factor increasing mortality in females. Although described here for the first time intra-specifically, this finding is consistent with the previous study[22] showing feeding inhibition of female *Ae. aegypti* by male *Ae. albopictus*. Interspecific mating of male *Ae. albopictus* with female *Ae. aegypti* actually occurs and is named satyrization[23,24]. Such feeding inhibition as well as mating disruption by male *Ae. albopictus* was also reported recently against *Ae. koreicus* females[25].

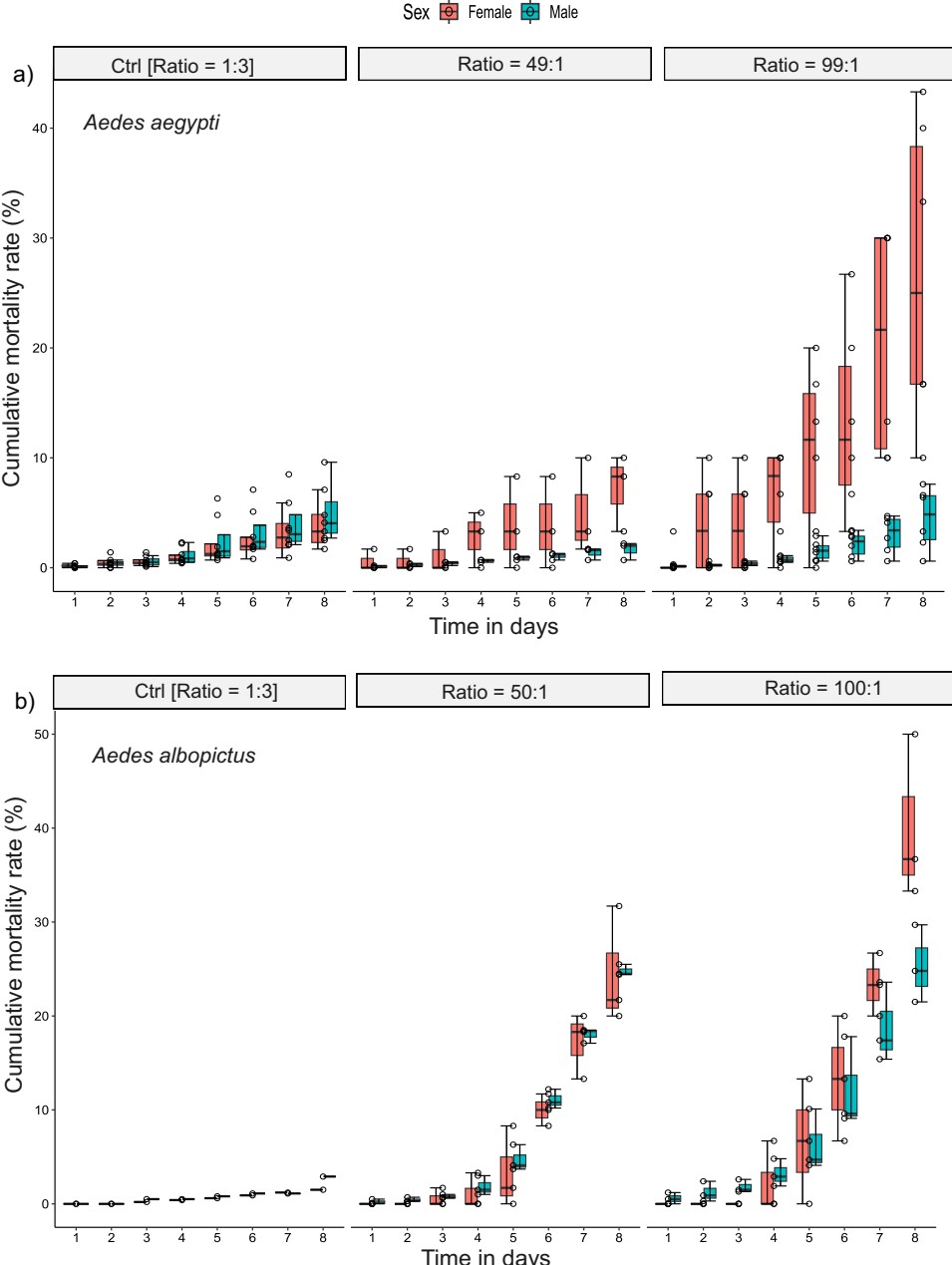

**Fig. 1 | Cumulative mortality rate of irradiated *Aedes* mosquitoes exposed to three sex ratio over 8 days.** The box plots present median values and quartiles, whiskers the 95% percentiles and dots the individual data points. One-tailed pairwise multiple comparisons were performed (*P* value adjustment with Tukey method) using the function *emmeans ()* of the emmeans package to investigate the significance of the increase in cumulative mortality rate at different sex ratios as compared to the control. **a** Cumulative mortality rate of *Ae. aegypti* females increased with sex ratio and was 26.7% ± 14.0% at 8 days for a ratio of 99:1 as compared to 3.9% ± 2.4% in the control group (odds ratio = 0.141, SE = 0.021, *z.ratio* = −13.128, *P* < 10⁻⁴). No difference was observed in males (odds ratio = 1.313, SE = 0.227, *z.ratio* = 1.575, *P* = 0.256). Significant difference was also observed in females with sex ratio 49:1 (odds ratio = 0.544, SE = 0.100, *z.ratio* = −3.289,

*P* = 0.002). The number of biologically independent replicates were *n* = 3 for the sex ratio 1:3 (control), *n* = 4 for ratio 49:1 and *n* = 6 for ratio 99:1; **b**, In *Ae. albopictus*, the tendency was even stronger. Significant difference was observed in females with sex ratio 49:1 (odds ratio = 0.106, SE = 0.049, *z.ratio* = −4.825, *P* < 10⁻⁴) and the cumulative mortality of females reached 40.0% ± 8.8% at 8 days for a ratio of 100:1 as compared to 3.8% in the control group (odds ratio = 0.106, SE = 0.049, *z.ratio* = −4.825, *P* < 10⁻⁴). Mortality of males also increased with the sex-ratio 50:1 (odds ratio = 0.093, SE = 0.039, *z.ratio* = −5.620, *P* < 10⁻⁴) and with the sex-ratio 100:1 (odds ratio = 0.081, SE = 0.034, *z.ratio* = −5.961, *P* < 10⁻⁴). The number of biologically independent replicates were *n* = 1 for the sex ratio 1:3 and *n* = 3 for ratio 50:1 and 100:1. Source data are provided in the Source Data file named "raw_-data_lab&semi-field.xlsx".

## Mating harassment and feeding success

In this experiment, we varied the male-to-female ratio while keeping the vector (female) to host ratio constant, in order to study the impact of mating harassment on host-vector contact.

We first explored the impact of a high irradiated males to non-irradiated female ratio on the feeding success of females on an artificial host (Hemotek). A male-to-female ratio of 99:1, reduced blood feeding

success to 1% (SE = 1%) as compared with 16% (SE = 4%) at a male-to-female ratio of 1:1 (odds ratio 16.50, SE = 9.98, *P* < 10⁻⁴) (Fig. 2a). Male mosquitoes were observed forming swarms around the artificial hosts waiting to mate with a female attempting to take a blood meal thus reducing their feeding success (see Supplementary Movie 3).

A similar experiment was set up but now using a human host. When a collector exposed one of his legs from foot to knee (human

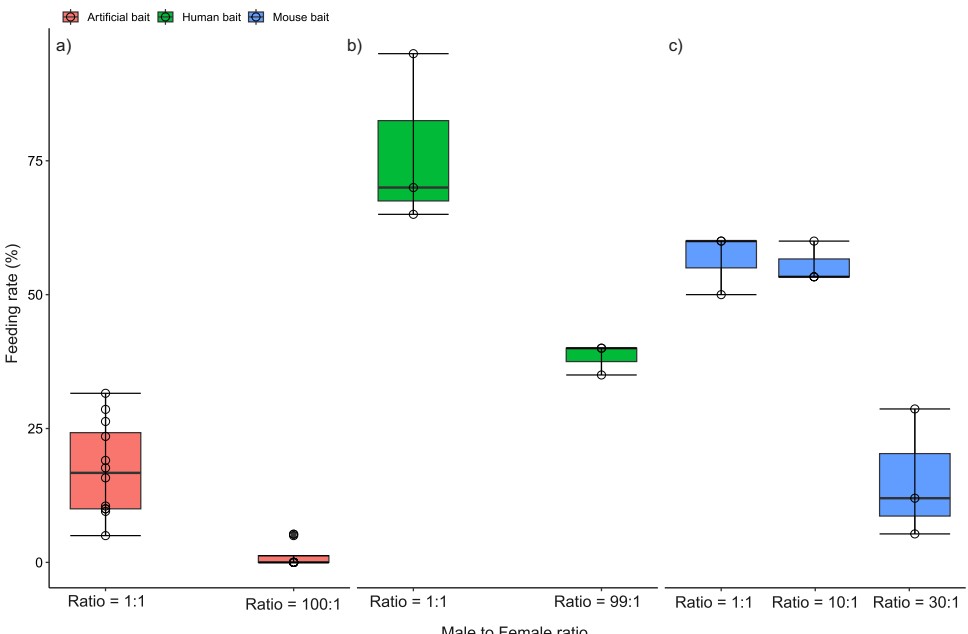

**Fig. 2 | Impact of mating harassment on feeding success in semi-field cages.** The box plots present median values and quartiles, whiskers the 95% percentiles and dots the individual data points. One-tailed pairwise multiple comparisons were performed (*P* value adjustment with Tukey method) using the function *emmeans ()* of the emmeans package to investigate the significance of the differences in feeding success at different sex ratios as compared to the control. **a** Impact of the male-to-female ratio on the engorgement rate of *Aedes aegypti* females on an artificial host (Hemotek). Fewer females were engorged in the male: female treatment ratio 99:1 as compared to the control ratio 1:1 (*n* = 12 biologically independent replicates, odds ratio 16.50, SE = 9.98, z.ratio = 4.641, *P* < 10⁻⁴). **b** Impact of the male-to-female ratio on the engorgement on the catch rate of

female *Aedes albopictus* by a volunteer collector. Fewer females were collected when attempting to bite a human collector in the male: female treatment ratio of 99:1 as compared to the control ratio 1:1 (*n* = 3 biologically independent replicates, odds ratio 5.30, SE = 2.15, z.ratio = 4.099, *P* < 10⁻⁴). **c** Impact of the male-to-female ratio on the engorgement rate of females on a mouse. Fewer females were collected when attempting to bite a human collector in the male: female treatment ratio of 30:1 as compared to the control ratio 1:1 (*n* = 3 biologically independent replicates, odds ratio 6.54, SE = 2.31, z.ratio = 5.306, *P* < 10⁻⁴) but no difference was observed between ratio 1:10 and 1:1 (odds ratio 1.05, SE = 0.314, z.ratio = 0.150, *P* = 0.987). Source data are provided in the Source Data file named "raw_data_lab&semi-field.xlsx".

bait) in a semi-field cage, and killed the female mosquitoes after landing on the exposed leg but before feeding began, the rate of caught females was reduced to 38% (SE = 6%) at a male to female ratio of 99 to 1 as compared to 77% (SE = 6%) with a male to female ratio of 1:1 (odds ratio 5.30, SE = 2.15, *P* < 10⁻⁴) (Fig. 2b).

Finally, a third experiment was conducted with an anesthetized mouse, where we used male-to-female ratios of 1:1, 10:1 and 30:1. We did not observe any reduction of the proportion of engorged females for a ratio of 10:1 (56%, SE = 5%) in comparison to the ratio of 1:1 (57%, SE = 5%). However, the feeding rate dropped to 17% (SE = 4%) at the male-to-female ratio of 30:1 (odds ratio 6.54, SE = 2.31, *P* < 10⁻⁴) (Fig. 2c).

In these three trials, mating harassment thus resulted in feeding inhibition, but only for high male-to-female ratios upon 30. Since aggregation of sterile males around human hosts during mosquito SIT programmes is well-known[7,26], we investigated if this might result in a similar feeding inhibition in field conditions.

**Mating harassment and human landing catches under field conditions**

The data from an *Ae. albopictus* field trial conducted in the centre of Guangzhou, China, were used to investigate the existence of feeding inhibition in real settings (Fig. 3). Before the release of sterile males, ovitraps were deployed bi-weekly in both the release and the untreated site to collect baseline data from March to August 2021 (Supplementary Fig. 6a). In addition, the density of the adult female populations was estimated with Human Landing Catch (HLC) (Supplementary Fig. 6b). Before the beginning of the release, no significant difference was observed in the number of hatched eggs per ovitrap and number of females caught with HLC in the untreated and release areas (Supplementary Fig. 6a, b). In addition, there was no significant difference

on the hatching rate of eggs between the untreated and release areas (mixed binomial model, *n* = 539, *z* = 1.684, *P* = 0.092). The density of the wild *Ae. albopictus* males was estimated to range from 6553 to 10,076 males/ha and from 2875 to 5292 males/ha via two independent performed mark-release-recapture experiments performed before the beginning of this trial (data not shown).

On 13th August 2021, the release of sterile male mosquitoes was initiated at a frequency of twice per week. During a period of 15 weeks, a total of 3 million male mosquitoes were released (Supplementary Fig. 6c). *Aedes albopictus* populations were monitored weekly with ovitraps, adult-collecting BG traps and irregular HLC. During the release period, the mosquito population was reduced in the release area by 47.56% and 35.96% as measured in the ovitraps based on the average of hatched eggs per ovitrap and the BG traps based on the average of captured wild females per BG trap, respectively, in comparison with the untreated area (Fig. 4a, b). From 6th September to 8th November, the efficiency of suppression was maintained at an average rate of 60.53% (min to max: 39.03%–86.07%) in the ovitrap catches. However, the suppression efficiency showed large variations after 8 November, and this might be attributed to the low ambient temperatures (12–22 °C) (Fig. 3c) or to possible immigration of fertile females in the release area in view of its small size (Fig. 3b, green line area, 1.17 ha). The temporal fluctuations of adult females were similar to the larval samples, i.e., an average suppression of 49.95% (min to max: 34.62%–92.50%) for the period 15 September to 2 December (excluding the data collected on 29 to 30 September) (Fig. 4b). After the beginning of the releases, we did not observe any significant impact on the hatch rate of eggs between the release (0.43, min to max: 0.32–0.52) and the control (0.44, min to max: 0.36–0.52) areas (mixed binomial model,

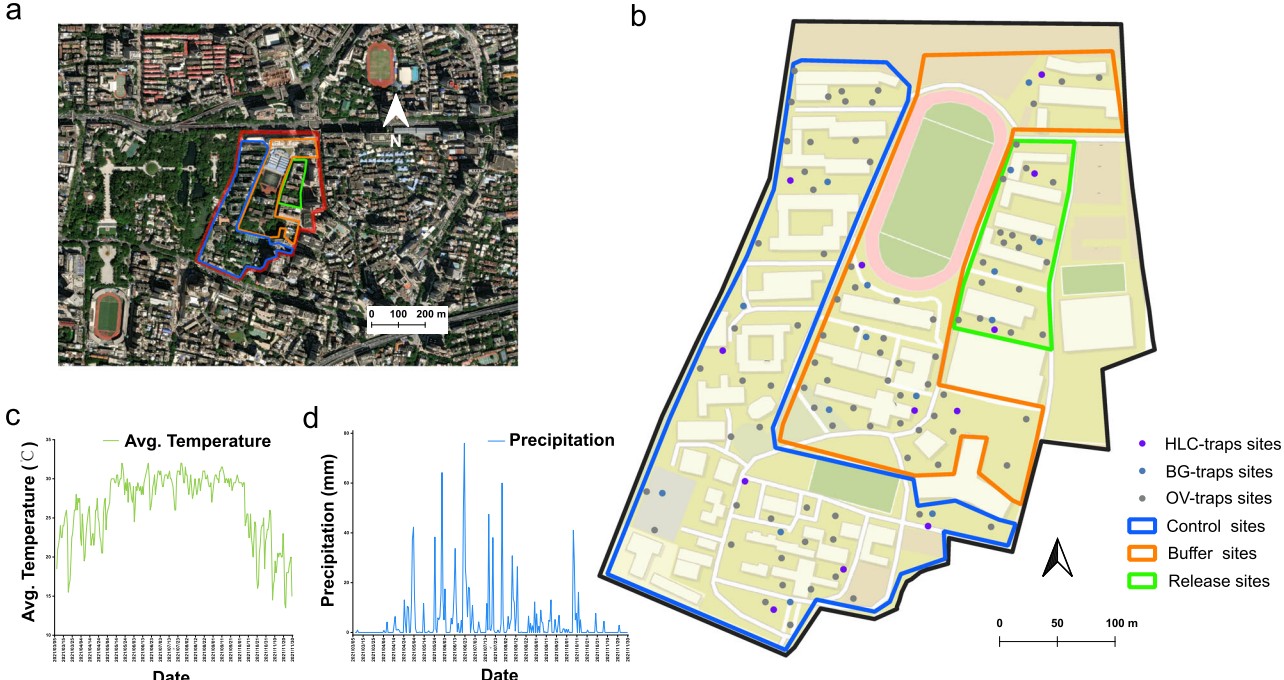

**Fig. 3 | Study site and climatic conditions. a** Satellite maps of field site in Guangzhou city (map data: Google, DigitalGlobal). Release area outlined with green while control and buffer areas are outlined with blue and orange in the satellite image respectively. N represents the North. **b** Spatial distribution of the monitoring tools/methods. Grey points represent ovitraps, blue points represent BG traps, and purple points represent the positions to perform Human Landing Catch. **c, d** Daily average temperature (**c**) and precipitation (**d**) in the study area from March to November 2021.

$n = 627$, $z = 1.129$, $P = 0.2591$), showing that induced sterility did not contribute to population suppression.

Eleven weeks after the first release of sterile males, we compared the sex ratio obtained by BG traps and HLC from 3rd to 6th November, and a higher sex ratio was found in HLC than in the BG traps (101.3:1 vs 12.5:1, Fig. 4c), although marginally significant because of strong variations between the three HLCs (min to max: 39.0 to 163.0). Quantitative polymerase chain reaction (qPCR) targeting *Wolbachia wsp* gene indicated that over 95% of caught males with BG traps or HLC were the released sterile males (Fig. 4d). In HLC, the sex ratio was close to the experimental set-up in our lab and semi-field studies presented above. An average of 0.50 adult females were collected in the release area versus 2.72 females in the untreated area using HLC. This indicated a mean suppression of 81.42% of adult females (Fig. 4e), a much higher suppression rate than what was observed with BG traps during the same period (42.31% during 3rd–4th November, Fig. 4b). The higher suppression rate obtained with the HLC might possibly be due to the high overflooding rate of males surrounding the catchers, which could have prevented the approach of female mosquitoes by the sterile males, as was observed in the semi-field trial. In *Aedes* species, males are known to swarm around the hosts using pheromonal and acoustic cues, presumably to intercept females attempting to feed[27–29]. Male *Ae. albopictus* are particularly attracted to humans[7] and our results show that they aggregated in higher numbers around humans than BG traps. At the same period, we did not observe a significant reduction of the number of adult females in the buffer area versus control area both in BG traps (3.40 ± 1.36, $n = 5$ and 4.33 ± 0.88 respectively, $n = 6$) and HLC (3.33 ± 0.60 and 2.72 ± 0.20 respectively, $n = 3$) despite a limited dispersal of sterile males outside the release area, which increased the sex ratio in the buffer area. In BG traps, the male-to-female ratio was 1.48 ± 0.71 ($n = 5$) in the buffer versus 0.44 ± 0.20 ($n = 6$) in the control area whereas it was 4.21 ± 1.10 versus 0.96 ± 0.28 respectively ($n = 3$) with HLC. This suggests that a strong male-to-female ratio is necessary to observe an impact of mating harassment in field conditions, as observed in the lab trial using mouse bait.

## Discussion

In various insect species, mating harassment is associated with costs that negatively affect the physical condition and hence, longevity of females, either through physical damage[30,31] or toxic effects from the accessory gland secretions[32,33]. In this study, however, females that were exposed to males at a 1:3 or 99:1 ratio and that were separated from the males immediately after the mating had a similar longevity (Supplementary Fig. 5), which was however shorter than for virgin females of the same age. This would indicate that during the exposure to male harassment, depletion of energy reserves and reduced feeding success were the main factors that reduced the longevity of mated females rather than any injury, as observed in other studies where reduced fertility was also documented[11,34]. Similar results were observed in other species when sex ratios were biased toward males, although to a lesser extent, like in the tsetse fly *G. morsitans morsitans*[35], in the dung fly *Sepsis cynipsea*[36], and the field cricket *Gryllus bimaculatus*[37]. Prevention of copulation by blocking or damaging the external genitalia of male tsetse flies resulted in reduced longevity of females caged with them, suggesting that the reduced female survival resulted from the physical aspects of male harassment rather than by components of the ejaculate[35]. In addition, male tsetse flies have a shorter lifespan due to being engaged in mating harassment of the females, as was likewise observed in our study in *Ae. albopictus*. Like in tsetse, *Ae. aegypti* female mortality was increased equally by caging them with males that had modified claspers to prevent mating or unmodified males[12]. These authors even suggest potential benefits (higher fitness) obtained from ejaculate components, a common phenomenon in insects that is considered as part of nuptial feeding[38].

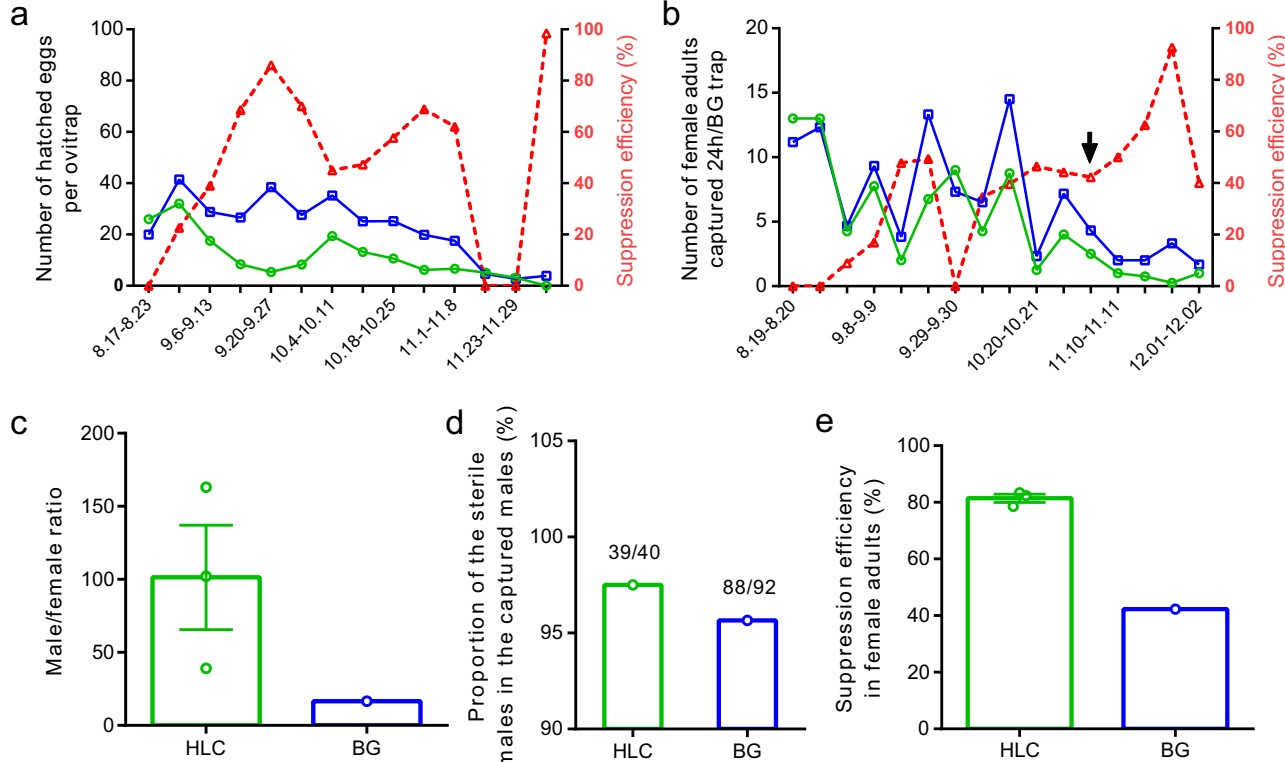

**Fig. 4 | Suppression efficiency of mosquito populations after sterile male releases. a** Dynamics of larval suppression. The release area is compared to the control area ($n$ = 14 samples, t = 4.209, df = 13, $P$ = 0.0010, Two-tailed Paired t test). b, Dynamics of adult female suppression. A total of 4 BG traps in the release area and 6 in the control area. Female reduction is observed in the release area ($n$ = 16 samples, t = 2.890, df = 15, $P$ = 0.0112, Two-tailed Paired t test). The red dotted lines indicate the suppression efficiency in both (**a**) and (**b**). **c** Ratio of males to females. An average ratio of 101.3 (±35.8) males to females was observed via HLC from 3rd to 6th November versus 12.5 via BG trapping on 3–4 November ($n$ = 3 samples, t = 2.367, df = 2, $P$ = 0.1415, One sample t test). **d** Proportion of sterile

males in the collected males via HLC and BG trapping. In both collecting methods, over 95% of collected males (HLC: 39/40; BG: 88/92) were sterile males, which were identified through qPCR based on the wsp gene of *Wolbachia*. The *Wolbachia*-negative samples were considered as the released sterile males. **e** Comparison on the suppression efficiency in adult females between HLC and BG trapping. Higher suppression efficiency was observed in HLC than in BG trapping (HLC: 81.42% ± 1.45%, $n$ = 3 samples; BG: 42.31%, as indicated by the black arrow in (**b**); $n$ = 3 samples, t = 26.95, df = 2, $P$ = 0.0014, One sample t test). All the data was presented as Mean ± SEM. Source data are provided in the Source Data file named "raw_data_field.xlsx".

The SIT is generally combined with other methods in an integrated pest management approach to first suppress the target population to a level low enough that sufficient sterile-to-wild male ratios can be obtained to induce enough sterility in the female wild population, e.g. in *Aedes* mosquitoes[6] or tsetse[39]. Hence, high sex ratios are not uncommon in SIT field trials. In operational tsetse fly SIT programmes, sterile to wild male ratios up to 100 were observed in some cases[39,40]. The sterile to wild male ratio peaked at 50 to 1 in another successful suppression program against *Ae. albopictus* in China[41]. One of the main benefits of the SIT is its inverse density-dependent properties[42] or in other words, the sterile to wild male ratio increases with each generation and with the rate of suppression and this can drive an insect population to extinction[40]. Our data show that feeding inhibition of the females might act synergistically to the induction of sterility in the female population. We did not observe any induced sterility in our field trial but the exact estimations of the hatch rates are not reliable because of a problem in the hatching method probably leading to an under-estimation of hatch rates (see Methods section). However, this problem (larvae eating part of the floating eggs) was independent from the treatment, suggesting that most of the observed suppression effect was related to other effects on females than induced sterility.

In both species studied here, feeding inhibition was demonstrated in the lab, together with direct impact on female suppression in the field in the case of *Ae. albopictus*. In addition to feeding inhibition, a possible explanation for increased female mortality in the field, is that

female-males aggregates may drop to the ground and/or attract predators[43]. Preliminary models have indeed predicted that an increase of efficacy of the sterile release programmes may be caused by male harassment[44]. However, such an impact will depend on the mating system of the target species[45] and it would be important to study this phenomenon in *Anopheles* or *Culex* species, or even *Ae. polynesiensis* that may use swarms triggered by visual cues instead or in addition to host-based mating.

Overall, our results allow us to propose two additional mechanisms contributing to the efficiency of the SIT against mosquito-borne diseases. First, we hypothesize that high male-to-female ratio increases female mortality through feeding inhibition thus directly reducing female lifespan. Second, at high male-to-female ratios, males reduce female feeding success and biting rate (and hence transmission rate). The SIT may thus directly reduce disease transmission at high male-to-female ratios through an impact on two critical components of vectorial capacity, namely female longevity and host contact[46]. This may as well occur in all genetic control methods based on inundative release of males, like the incompatible insect technique[41,47] or RIDL[48] or even those driving maleness into wild populations[49]. These hypotheses warrant more field research to assess the impact of these mechanisms on disease transmission.

## Methods

This research complies with all relevant ethical regulations. For the study involving Human Landing Catch in large cages, the protocol was

approved by the Ethics Committee on Laboratory Animal Care of the Zhongshan School of Medicine (ZSSOM), Sun Yat-sen University (No. 2018-020). The experiment involving the use of anesthetized mice to blood-feed *Aedes* mosquitoes was conducted according to protocols on Laboratory Animal Care approved by ZSSOM (03/14-036-00 and No. 2017-041). The field trial on applying SIT for *Ae. albopictus* control has also been reported to and approved by ZSSOM before the release of sterile males in 2021.

## Impact of male harassment on mortality in laboratory trials

All experiments on *Ae. aegypti* were carried out at the Insect Pest Control Laboratory (ICPL, IAEA, Vienna, Austria, whereas experiments on *Ae. albopictus* were conducted independently at IRD, Saint-Denis, La Reunion Island, except one preliminary experiment on *Ae. albopictus* also conducted at IPCL (Supplementary Fig. 4). The objective of organizing studies among different locations and with different strains was to strengthen the observations, but only *Ae. albopictus* was available in Reunion.

**Mosquito colonies and mass-rearing.** Three established mosquito colonies of *Ae. aegypti* and *Ae. albopictus* were used to perform these experiments.

At the IPCL, the strain of *Ae. aegypti* and *Ae. albopictus* originated respectively from Juazeiro, Brazil in 2012 (provided by Biofabrica Moscamed, IAEA Collaborative Center) and Rimini, Italy in 2018 (provided by Centro Agricoltura Ambiente, IAEA Collaborative Center). These two strains were maintained at the IPCL in a 264 m² container-based laboratory under controlled environmental conditions: the larval rearing room was maintained at 28 ± 2 °C, 80 ± 10% RH and the adult rearing room at 26 ± 2 °C, 60 ± 10% RH, with a 14:10 h light: dark (L:D) photoperiod with 1-h periods of simulated dawn and dusk in both rooms. *Aedes* mosquito eggs older than 2 weeks were obtained from mass-rearing procedures developed at the IPCL[50,51]. Based on the egg hatch rate calculated from sub-samples of 100 eggs, batches of eggs corresponding to approximately 18,000 first instar larval (L1) were estimated following the method described by Zheng, et al.[52], weighed and then hatched separately in glass jam jars filled with 700 mL of boiled and cooled reverse osmosis water with the addition of 10 mL of larval FAO/IAEA diet[50,53]. The larvae were reared into mass-rearing trays following the mass- rearing procedures developed by the IPCL[50]. Larvae were reared with larval diet (4% w/v) composed of a combination of powdered tuna meal (50%), black soldier fly (35%) and brewer's yeast (15%).

At IRD, Saint-Denis, Reunion Island, the strain of *Ae. albopictus* used in the experiments originated from Saint-Benoit, Reunion Island and was reared as adults in a climate-controlled room maintained at a temperature of 27 ± 2 °C and 75 ± 1% relative humidity; the light regime was LD 12:12 h photoperiod. For larval production, batches of four thousand first instar larvae were counted on day 0 into rearing trays (52 × 32 × 6 cm) containing 2 L of tap water. Larvae were reared at a room temperature of 31 °C and a photoperiod of 12:12 (L:D) and fed with 10, 20, 25, 25 and 20 ml per tray of a solution at 7.5% (w:v) slurry of diet (50% ground rabbit-food and 50% ground fish-food Tetramin, Tetra, Germany) on days 0, 1, 2, 3 and 4, respectively. Pupae appeared from the fifth to the seventh day.

**Experimental design.** At the IPCL, *Ae. aegypti* pupae were sexed mechanically using a Fay-Morlan glass plate separator[54] as redesigned by Focks (John W. Hock Co., Gainesville, FL)[55]. With this method, the female contamination in males collected on the first tilting is generally 1.11 ± 0.27% on the first day of tilting[56].

Additionally, samples of sorted male pupae were checked under a binocular microscope to measure the desired ratio. Batches of 3000 male and female pupae were counted and left to emerge inside separate Bugdorm cages (30 × 30 × 30 cm). Throughout emergence, the cages were monitored to remove females from the male batches and

males from the female batches to achieve complete male and female separation. Adults were maintained with 10% sucrose solution supplied *ad libitum* in a 150 mL plastic cup containing a sponge.

To study the sexual harassment of males on *Aedes* mosquito females without allowing potential density-dependent mortality, batches of 3000 *Ae. aegypti* mosquitoes aged 0 to 1 day were placed in the Bugdorm cages (30 × 30 × 30 cm) at six male-to-female sex ratios (SR): SR = 3:7, SR = 1:3 (control, used in the colony maintained in mass-rearing conditions), SR = 10:1, SR = 23:2, SR = 49:1 and SR = 99:1. Every day at 10 am, these cages were monitored and mortality was recorded during 8 days (13 days in the preliminary trials on non-irradiated males). During preliminary trials, the cumulative mortality rate of females increased for batches with SR of 10:1; 23:2; 49:1; and 99:1 after eight days. It was thus decided to focus on SR of 99:1 and 49:1 to study the effects of sexual harassment in sterilized *Aedes* mosquitoes. Batches of sterile mosquitoes with a SR of 1:3 were again used as controls. Furthermore, one trial was organized for the *Ae. albopictus* Rimini strain using a SR of 99:1 and a SR of 1:3 as control (Supplementary Fig. 4). The batches of *Ae. aegypti* (both sexes) were irradiated at the adult stage at 60 Gy while the batches of *Ae. albopictus* at the pupal stage at 40 Gy.

To assess the longevity of harassed females after separation from the males, 20 irradiated females from batches at the SR of 99:1, 20 irradiated females from batches at the SR of 1:3 and 20 irradiated virgin females of the same age were placed into Bugdorm cages (15 × 15 × 15 cm, Taiwan, China) at the IPCL. Mortality checks were carried out daily over 14 days.

At IRD, Saint-Denis, Reunion Island, *Ae. albopictus* pupae were sexed mechanically using standard metal sieves with a square-opening mesh through which males swim upward[57].

After sex separation, male pupae were allowed to emerge into Bugdorm cages (30 × 30 × 30 cm) with constant access to a 5% sucrose solution [w/v]. Female pupae were first isolated in tubes (5 per tube) to check the accuracy of the sexing at the emergence and then transferred into cages already containing males. Two treatments were repeated three times, a ratio of 100:1 (male: female) and a ratio of 50:1 with 3000 males and 30 females and 3000 males and 60 females respectively, in Bugdorm cages (30 × 30 × 30 cm) with constant access to a 5% sucrose solution. Control cages consisted of regular rearing cages with a ratio of 1: 3 (male: female). Each treatment has been done with non-irradiated and irradiated males. Mortality checks were carried out daily and recorded over 8 days.

To produce irradiated males, male and female pupae of more than 30-h-old were irradiated at 35 Gy during 5 min with an X-ray irradiator (Blood X-RAD 13–19, Cegelec, France) at the Blood bank coordinated by Etablissement Français du Sang (EFS) located at the Bellepierre hospital, St Denis de La Réunion. The irradiated pupae were brought back to the lab and treated as described above.

## Mosquito recordings

These recordings were implemented in Singapore because they require specialized equipment and unfortunately, only *Ae. aegypti* was available. The strain of *Ae. aegypti* used for filming originated from Singapore and reared at the National Environment Agency–Environmental Health Institute (NEA-EHI) Singapore, mosquito production facility. The larvae were reared at a High Density Mosquito Rearing System (Orinno Technology, Singapore) at larvae density of 12,000 per tray containing 6 litres of water and maintained at an air temperature of 29 ± 1 °C and 85 ± 5% RH with a photoperiod of 12:12 h L:D cycle. *Aedes aegypti* pupae were sexed mechanically using an Auto-Pupae Separation System (Orinno Technology, Singapore). Male and female pupae were placed into two separated Bugdorm cages (30 × 30 × 30 cm) to allow emergence. Adults were supplied with 10% sucrose solution *ad libitum*. Adult mosquitoes with age of 5–6 days postemergence were selected for filming via mouth aspirator.

All footages were recorded by DJI OSMO pocket and Nikon D750 DSLR camera with Sigma 70 mm F2.8 Macro lens. Two Yongnuo YN900 LED panel lights were used as light source. For Supplementary Movie 1, two female *Ae. aegypti* adults were introduced into a Bugdorm cage (30 × 30 × 30 cm) with 200 males. Footage was captured by manual tracking at 60 Frame Per Second (FPS) and down speed to 30FPS in the postediting. For Supplementary Movie 2, a single female was knocked down by exposure to ethyl acetate and carefully sticked to the head of pin with latex glue. The immobilized female was then placed into a Bugdorm cage (30 × 30 × 30 cm) with an additional 100 males for filming. Footage was captured at frame rate of 30 FPS.

## Feeding inhibition trials

### Artificial bait (Austria)

**Mosquito strains, rearing, and irradiation.** Two mosquito laboratory strains of *Ae. aegypti* (FAO/IAEA, 2017, 2020) were used for these experiments. The strains were maintained following FAO-IAEA guidelines[58]. *Aedes aegypti* strains originating from Brazil (Juazeiro) and Senegal (Dakar) were transferred to the IPCL from the insectary of Biofabrica Moscamed, Juazeiro, Brazil, and from the ISRA-LNERV, Dakar-Hann, Senegal in 2012 and 2021, respectively.

The larval rearing period had controlled conditions of temperature of $28 \pm 2\,°C$, $80 \pm 10\%$ RH, and lighting of 14:10 h L:D, including 1 h of dawn lighting and 1 h of dusk lighting for larval stages. Adults were separately maintained under $26 \pm 2\,°C$, $60 \pm 10\%$ RH, and 14:10 h light: dark, including 1 h dawn and 1 h dusk. To perform the experiments, mosquitoes were reared following modified mass-rearing procedures developed at the IPCL[59]. Pupae were collected and mechanically sex-separated using a semi-automatic pupal sex sorter (Wolbaki, China).

Pupae were counted manually and placed in 30 × 30 × 30 cm and 15 × 15 × 15 cm Bugdorm cages for male and female mosquitoes, respectively. Pupae were aliquoted into 600 mL plastic cups, each holding 2100 male pupae and into 100 mL plastic cups (Medi-Inn, United Kingdom) each holding 25 female pupae. Adults were maintained with *ad libitum* access to a 10% (w/v) sucrose solution until the day of the irradiation. Mortality was assessed daily until the day of releases.

Two-to-three–day-old male adults were exposed to 45 Gy using an X-ray blood irradiator (Raycell MK2)[60]. Male adult mosquitoes were held in a cold room at $4\,°C$ for ten min in compacted batches of 100/ cm$^3$ (about 1000 males /cell) to simulate mass-transport conditions prior to irradiation. Irradiated male mosquitoes were placed back into the cages with *ad libitum* access to a 10% (w/v) sucrose solution until testing day in the Ecosphere of the FAO-IAEA Insect Pest Control Laboratory (Supplementary Movie 3). Approximately 24 h prior to the releases, female mosquitoes were starved by removing the sugar solution from all cages. Two ratios of males to virgin females of 99:1 (1980:20) and the control ratio 1:1 (20:20) were used with three cages each (technical repeats).

**Sexual harassment assay in large cages.** Experiments were conducted in six large cages (1.80 × 1.80 × 1.80 m, Live Monarch, Boca Raton, USA) at the FAO-IAEA IPCL climate-controlled Ecosphere in Seibersdorf (Austria) under natural light, average temperatures of $28 \pm 2\,°C$ and $70 \pm 10\%$ RH (Supplementary Movie 3). One tray (30 × 40 × 8 cm) containing 1 L tap water was provided in each cage with two 100 mL plastic cups of 10% sugar solution. A stand made of wood was placed inside each cage to hold an Hemotek (Ltd Unit 5 Union Court Great Harwood Business Zone Blackburn BB6 7FD, United Kingdom) blood feeding plate[61] as artificial bait. One blooding plate was filled up with 100 mL fresh pig blood and was hung upside down. The Hemotek heating system was turned on for 30 min. The plate was placed halfway of the wooden stand at one meter above the floor and allowed females to feed easily.

Five-to-six-day-old, irradiated males and virgin non-treated female mosquitoes were briefly knocked down for five to ten minutes at $4\,°C$ prior to release. Mosquitoes were then transferred into 100 mL plastic containers. Each container was labelled according to treatment or control groups. All the containers were then transferred to the Ecosphere and males were released into large cages. Females were released 30 min later where they were allowed to blood feed for two hours starting from 10:00 am.

After 2h-exposure time, all females were recaptured separately from the treatment and the control cages using mechanical aspirator device[62]. The operator wore coverall protective suit and gloves preventing any biting from the females during collection. The number of recaptured females was recorded per cage. To assess the blood-feeding status of females, each recaptured female mosquito was crushed between two pieces of white paper and the visual presence/ absence of blood was observed based on the blood stain. The number of blood-fed females was recorded per cage. In total, three replicates (cage) were prepared for the control sex ratio (males: virgin females) of 1:1 (20:20) and the treatment sex ratio of 99:1 (1980:20). The full experiment was repeated four times.

### Human bait (China)

**Mosquito strains, rearing, and irradiation.** The female mosquito GUA line was collected from more than 10 field localities of Guangzhou City, China, and has been reared in the laboratory for less than one year (<12 generations). The rearing conditions for GUA were described previously[63]. Briefly, about 300 first-instar larvae were reared in a plastic tray (L*W*H = 36 × 25 × 5 cm) with 1.5 L dH$_2$O and bovine liver powder was supplied as larvae food. The HC line was generated through *Wolbachia* embryonic transinfection, and contains three *Wolbachia* strains: the two naturally occurring strains in *Ae. albopictus* (*w*AlbA and *w*AlbB) and a newly acquired strain from *Culex molestus* (*w*Pip)[41]. For egg mass production, approximately 3000 females and 1000 males were placed in a 30 × 30 × 30 cm cage. A 10% sucrose solution was provided daily as food resource, while the females were fed with commercial sheep blood (Future Biology, China) for egg production. After egg maturation, eggs were hatched and approximately 7000 larvae were reared in a tray (L*W*H = 58 × 38 × 4 cm). The larvae were fed a daily diet consisting of 60% liver powder, 30% shrimp powder, and 10% yeast. After pupation, male and female pupae were separated using a Fay-Morlan sorter. Male pupae were subsequently placed in a canister (7.5 cm in diameter and 7.5 cm in height) and exposed to 45 Gy irradiation. Then approximately 3000 male pupae were placed in a 30 × 30 × 30 cm cage for eclosion. The resulting male adults were utilized for the human landing catch experiments in large cages.

**Human Landing Catch in large cages.** We conducted a second experiment based on Human Landing Catch in China to assess whether male harassment can prevent blood feeding on humans in semi-field conditions. Wild type virgin *Ae. albopictus* (GUA strain) females were inseminated at 5–6 days old. They were starved for 24 h before the experiment start. Irradiated HC males were virgin and 5–6 days old. Irradiated HC males were released into semi-field cages (1.80 × 1.80 × 1.80 m, containing two sugar water containers). GUA females were released 24 h later into the semi-field cages. Male and female release numbers were 1980 versus 20 for the 99:1 ratio and 20 versus 20 for the 1:1 ratio. The mosquitoes were immobilized by placing them in a chilling room with a temperature of $8 \pm 2\,°C$. Subsequently, we conducted a precise count of the required number of mosquitoes. Ten minutes after releasing the females, an adult volunteer wearing long-sleeved shirt, long pants, gloves, and mosquito-proof hat entered and sat on a chair in the middle of each cage. The collector exposed one of his legs from foot to knee and killed mosquitoes as soon as they landed on the exposed leg before they started feeding. Since there were

female mosquitoes attempting to feed on the volunteers but failed due to mating harassment from males, only the females that successfully landed on exposed skin were classified as potential "successful bloodsuckers". To avoid the collection of male mosquitoes and "unsuccessful bloodsuckers", we opted to eliminate the landed females by swatting and killing them rather than using an aspirator. This approach was necessary due to the large number of male mosquitoes forming a swarm around the volunteer and the presence of female mosquitoes facing harassment while attempting to feed. This method prevented the collection of non-target mosquitoes. Mosquito collection was conducted for 15 min for each cage and ratio. All collected females were removed and counted. After 15 min of collection, remaining mosquitoes were collected with an aspirator and females individually checked for blood feeding or not. Three repeats were conducted with three different collectors managing one 99:1 and one 1:1 cage each. Collectors received appropriate information and gave their informed consent prior to participating in this study.

**Mouse bait (China).** Thirty 3–4-day-old virgin female mosquitoes were placed in a 30 × 30 × 30 cm Bugdorm cages, after 30, 100 and 900 males were introduced into the cage, corresponding of SR of 1:1, 10:1 and 30:1, respectively. Each SR was repeated in 3 cages. Two days later, female mosquitoes were fed on an anesthetized mouse for 1 h. Following this exposure, all mosquitoes were removed out of cage with a fan-aspirator and then anesthetized with $CO_2$. The total number of residual females and the number of engorged females were recorded.

### Field trial

The field trial was organized against *Ae. albopictus* only because we took the opportunity of a preliminary SIT trial organized in China against this species, offering perfect settings to investigate the impact of male mating harassment in the absence of a strong induced sterility component, particularly a small and non-isolated release area.

**Maintenance of mosquitoes.** We used the *Ae. albopictus* GT line (without *Wolbachia* infection) that can be distinguished from the wild *Ae. albopictus* (*w*AlbA and *w*AlbB double infections) via PCR/qPCR assays based on *Wolbachia wsp* gene (see sequence below). Briefly, the GT line was generated from the wild GUA line via providing the GUA line with 10% sugar solution containing tetracycline (1.0 mg/mL) for five consecutive generations and another two generations without tetracycline. The GT line was maintained on 10% sugar solution at 27 ± 2 °C and 75 ± 10% humidity with a 12-h light/dark cycle, according to standard rearing procedures[64].

**Mass production and irradiation of GT males.** Mass production of GT males included adult and larval rearing according to protocols described previously with slight modifications[18,65]. Briefly, approximately 15,000 female pupae and 5000 male pupae (3:1 ratio of female to male) were placed into an adult cage (90 × 90 × 30 cm). Adults were provided with a 10% sugar solution *ad libitum*. Sheep blood mixed with ATP was provided to females twice per rearing cycle. Oviposition cup was provided to the engorged females for laying eggs 48 h after each blood meal. Eggs were collected for 72 h and then matured for at least one week before hatching. After hatching, 4000–5000 larvae were added to each tray (51.5 × 36.0 × 5.5 cm) and fed daily with larval food. At day 8, pupae mixed with larvae were collected and then separated by an automatic sex separator (Orinno Technology, Singapore). After sexing, 16,000 male pupae were transferred to a cage (90 × 90 × 30 cm) for emergence. The temperature was set at 27–28 °C. Cotton soaked in 10% sugar solution was placed on top of the cage for mosquitoes to feed *ad libitum*. The average female contamination rate was 0.05% ($n = 30$, SE = 0.02%) in the sterile male release batches (Supplementary Fig. 6c). Male mosquitoes at 2–3-day old were immobilized

and then packed in plastic dishes (diameter 10 cm × height 1.2 cm) in a cooling room set at 10 °C. Each plastic dish contained 5000 male mosquitoes and was then placed in a PMMB canister. Each irradiated canister contained 3 dishes and two canisters were irradiated each time. The exposure was done in an X-ray irradiator (XL1606HD, NUCTECH, China) at a dose of 60 Gy with dose rates of 3.74 Gy/min or 7.33 Gy/min. The irradiator was configured with a cooling system to maintain the chamber temperature at 10 °C, which ensured the immobilization of male mosquitoes during exposure without impacting their quality[66–68]. The irradiated male mosquitoes were recorded as $IGT_{60Gy}$ males. Exposing adult male mosquitoes to 60 Gy resulted in an average of 99.0% sterility ($n = 30$, SE = 0.22%, Supplementary Fig. 6c).

**Quality control.** One of the key quality control parameters for release of sterile males was the female contamination rate (FCR), which was monitored at the adult stage. Each batch of male adults was checked by randomly selecting 800–1000 of the mosquitoes for sex identification based on morphology. In addition, male sterility was monitored for each batch through egg hatch rate assessment. In details, 100 $IGT_{60Gy}$ males were allowed to mate with 100 virgin GT females. Blood feedings and egg collections were the same as mentioned above. Eggs from each blood meal were hatched and egg hatch rate was assessed under the stereomicroscope by counting the number of eggs hatched and the total egg number[64]. Egg hatch rate from crosses between 100 GT males and 100 virgin GT females was considered as fertile control. Male sterility was calculated as: Induced Sterility (IS%) = 100% - ((Hs/Hn) * 100%), where Hs was the egg hatch rate from the sterile control, and Hn was the egg hatch rate from the fertile control.

**Study area description.** The study site is located at the North Campus of Sun Yat-Sen University in Yuexiu District, Guangzhou, China (Latitude: 23°7′39.74″N, Longitude: 113°17′22.07″E), covering an area of about 20.9 ha (Fig. 3a). The campus has a population of 4750 people (mainly students and faculty) and is located in a bustling metropolitan area with parks, hospitals, and residential areas nearby. The west and south areas of the campus were selected as the control area (6.55 ha), the northeast was the release area (1.17 ha), and a buffer zone (4.87 ha) was set between the release and the control area (Fig. 3b). The average temperature in the study area was 24.6 °C in 2021 (Fig. 3c) and the annual precipitation was 1511.4 mm with a rainy season between May and October (Fig. 3d).

**Pre-release monitoring of release and control areas.** Before release, *Ae. albopictus* populations were monitored using ovitraps every two weeks from 8th March to 17th August 2021. The number of ovitraps was 17 in the release area, 40 in the control area and 33 in the buffer area, respectively (Fig. 3b). The methods to place and collect ovitraps, as well as hatch eggs, were the same as described in[41] with some modifications. Briefly, the ovitrap constituted of transparent bottles with a black lid with three holes, allowing engorged females entering the trap for laying eggs. The ovitraps were cylindrical plastic containers of 70–75 mm diameter and 100 mm height. Before using, a piece of filter paper (70 mm width and 45 mm height) was inserted along the ovitrap wall. A 50 mL bamboo leaf solution was added in the ovitrap to increase the trapping efficiency. Ovitraps were placed close to the natural breeding sites of *Ae. albopictus* for 7 days. The positive ovitraps were collected and incubated for another 7 days at room temperature before counting the number of eggs and the hatched larvae. Positive traps where eggs or larvae were observed were selected for further evaluation. The filter paper with eggs were removed from the ovitrap and the egg hatch rate was determined under a stereomicroscope. Boiling water was used to kill the remaining larvae and the number of larvae was counted. Unfortunately, we observed that some of early-laid eggs hatched before initiating the hatching procedure and that larvae

ate some of the floating eggs, which resulted in more larvae than eggs for some time-point data (see raw data). We thus considered a hatch rate of 1 for all these data points in the analysis.

We also performed Human Landing Catch (HLC) to estimate the mosquito adult populations. There were two positions in the release area and 6 positions in the control area (Fig. 3b). The HLC was performed 4 times pre-release of sterile males. Briefly, well-protected volunteers stand in the selected position and used a locally manufactured hand-held electric aspirator to collect the adult mosquitoes flying around the performers for 15 mins. The collected mosquitoes were identified and counted by morphological characteristics.

**Field release of IGT$_{60Gy}$ males.** IGT$_{60Gy}$ males were maintained in a mobile refrigerator set at 10 °C and transported from the mass-rearing factory to the study site by a van two times per week. The distance between the factory and the study site was about 100 km. The release was performed at 13:00–14:00 pm. During release, dishes were opened, and mosquitoes were allowed to fly away freely. Over 95% of mosquitoes could recover after transportation under chilling conditions. On average, 200,000 mosquitoes were released weekly, and a total of about 3-million mosquitoes were released from mid-August to end of November 2021.

**Monitoring population suppression.** Throughout the period of IGT$_{60Gy}$ male release, *Ae. albopictus* populations were monitored weekly by using ovitraps and BG-Sentinel traps (Biogents, Germany). The number of BG traps was 4 in the release area, 6 in the control area and 5 in the buffer area (Fig. 3b). The BG traps were generally placed under trees or in the bushes, avoiding sunlight and heavy raining. Each BG trap was supplied by a 12.2AH lead-acid battery (6-DZF, Tianneng, China) and kept capturing for 24 h once per week. The collected mosquitoes were killed, identified for both species and sex, and the number was recorded, respectively. The average number of hatched eggs per ovitrap, in both release and control areas, was determined and used to measure population suppression efficiency at the larval stage. In addition, the average number of females in both release and control areas per BG trap was determined each week, and used to measure population suppression at the adult stage. Moreover, HLC was repeated three times to estimate the suppression efficiency at 11 weeks' postrelease of IGT$_{60Gy}$ males.

**qPCR assays of *Wolbachia* infection.** Each captured adult mosquito was stored separately in a 1.5 mL tube and maintained at −20 °C before *Wolbachia* detection. DNA was extracted according to the protocols of Fast Pure Cell/Tissue DNA Isolation Mini Kit (DC102-01, Vazyme, China). A 20 μL qPCR reaction consisted of 1 μl DNA template, 10 μL ChamQ Universal SYBR qPCR 2X mix (Vazyme, China), 8 μL nucleic-free water, 0.5 μL primer-F, and 0.5 μL primer-R. The specific primers used for the assay were designed for *Wolbachia wsp* gene and consisted of *w*AlbB-F: ACGTTGGTGGTGCAACATTTG; *w*AlbB-R: TAACGAGCAC-CAGCATAAAGC. The qPCR procedures (LightCycler 96, Roche) comprised 10 s at 95 °C, followed by 40 cycles of 10 s at 95 °C, 10 s at 50 °C, 10 s at 72 °C, and finally 10 s at 95 °C, 60 s at 65 °C, 1 s at 97 °C, 30 s in 37 °C to generate the melting curve for confirmation that the fluorescence detected was for the specific PCR product. The *Wolbachia*-negative samples were considered as IGT$_{60Gy}$ mosquitoes.

**Statistical analysis**
All statistical analyses were performed using R version 4.2.1 (https://cran.r-project.org) using RStudio 2022.07.1 (RStudio, Inc. Boston, MA, United States, 2016). Shapiro and Bartlett's tests were performed respectively to test the normality and to determine whether the variance in cumulative mortalities was the same for various sex ratios. The relationships between cumulative mortalities and the different sex ratios during the study period were analysed for each

*Aedes* species. For this purpose, binomial linear mixed effect models were used with the assigned sex ratios as response variables and cumulative mortality rates as explanatory variable using the lme4 package[69]. The various sex ratios were then used as fixed effects and the repetitions as random effects. Binomial linear mixed effect models were also used to analyse the impact of the releases on the hatching rate, with the time after the beginning of the releases, the treatment (release, buffer, control) and their first order interaction as fixed effects, and traps id and dates as random effects. The generalized linear mixed models were fitted by maximum likelihood. For each species, the cumulative mortality curves were plotted by sex ratios using the ggpubr package. The longevity of harassed, non-harassed and virgin *Ae. aegypti* females was analyzed using Kaplan-Meier survival analyses. The log-rank test (Mantel-Cox) was used to compare the level of survival between the different treatments (status of females) using the survival and survminer packages[70]. Two-tailed Paired t test was used to compare the hatched eggs and the captured female adults via BG or HLC before and after the release of sterile males, between the released and control areas. The feeding rates and recapture rates of females in semi-field trials were analysed using binomial generalized linear mixed models fit by maximum likelihood (Laplace approximation) with the SR as fix factor and the repeats as random factors[71]. The odds ratio were computed using the emmeans function (in package emmeans)[72].

**Reporting summary**
Further information on research design is available in the Nature Portfolio Reporting Summary linked to this article.

## Data availability
The raw data generated by this study and used as source data of all Figures and Supplementary Figs. are provided as a Source Data file that is publically available (CC BY 4.0, https://doi.org/10.6084/m9.figshare.23578995).

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

## Acknowledgements

This research was funded by the United States of America under a grant of the IAEA entitled "Surge expansion for the sterile insect technique to control mosquito populations that transmit the Zika virus" (JB). Sun Yat-sen University was funded by the National Key Research and Development Program of China (2020YFC1200100 and 2022YFC2603600), the National Natural Science Foundation of China (82002168 and 82072308), the 6th Nuclear Energy R&D Project (20201192), the Science and Technology Planning Project of Guangdong Province, China (2021B1212040017 and 2022B1111010004), the Guangzhou Basic and Applied Basic Research Foundation (202201011518), the IAEA Department of Technical Cooperation (RAS5095), the IAEA Coordinated Research Project (D44005), the NSFC-BMGF (82261128006 and 2022YFML1005) and BMGF (INV-061480) (DZ). This research was also part of the Coordinated Research Project (CRP) of the FAO/IAEA Centre of Nuclear Techniques in Food and Agriculture on irradiation and quality control (MTB). Wolbaki Biotech was funded by the Guangdong Innovative Research Team Program (No. 2011S009) (ZX). IRD was funded by the ERDF program, grant number "GURDTI 2017-0583-0001899" within 2014–2020 framework—Action 1.05 "Strengthening the Health and biotechnology Innovation" (DD). The funders had no role in the design of the study; in the collection, analyses or interpretation of data; in the writing of the manuscript or in the decision to publish the results.

## Author contributions

J. Bouyer, D.Z., H.M. and Z.X. developed the concept and methodology; H. Maiga, M.T.B., D.D., W. M., N.S.B.S., T.W., O.B.M., C.M. and S.S.K performed the lab experiments; Y. Li, H.M., W.M., N.S.B.S. and H.Y. performed the semi-field experiments; D. Zhang, G.W., Y.S., J.G., Q.F., J.Z., X. Zhao, D.P., W.P., Y.W., X. Zheng, and Z.W. performed the field trial, D. Lu, C.H.T. and J.B. performed the movies; J. Bouyer, D.Z., C.H.T., Y.W., Z.X. and M.J.B.V. performed coordination for the project; D. Zhang obtained regulatory approvals for mosquito releases; Z. Xi obtained the ethical permit for the semi-field trial involving human bait; J. Bouyer provided oversight of the project and contributed to all experimental designs, data analysis and data interpretation; J. Bouyer, D.Z., H.M., Y.L., D.D., C.M., D.L., Z.X. and M.J.B.V. wrote the manuscript. All authors participated in manuscript editing and final approval.

## Competing interests

Y. Li and Z. Xi are affiliated with Guangzhou Wolbaki Biotech Co., Ltd. W.P. is affiliated with SYSU Nuclear and Insect Biotechnology Co., Ltd. The other authors declare no competing interests.

## Additional information

[1]Chinese Atomic Energy Agency Center of Excellence on Nuclear Technology Applications for Insect Control, Key Laboratory of Tropical Disease Control of the Ministry of Education, Sun Yat-sen University, Guangzhou, China. [2]Insect Pest Control Laboratory, Joint FAO/IAEA Centre of Nuclear Techniques in Food and Agriculture, IAEA, Vienna, Austria. [3]Institut de Recherche en Sciences de la Santé, Direction Régionale de l'Ouest (IRSS-DRO), Bobo-Dioulasso, Burkina Faso. [4]Department of Pathogen Biology, School of Medicine, Jinan University, Guangzhou, China. [5]Guangzhou Wolbaki Biotech Co., Ltd, Guangzhou, China. [6]Institut Sénégalais de Recherches Agricoles, Laboratoire National de l'Elevage et de Recherches Vétérinaires, BP 2057 Dakar, Sénégal. [7]Institut de Recherche pour le Développement (IRD), UMR MIVEGEC (CNRS/IRD/Université de Montpellier), IRD Réunion/GIP CYROI (Recherche Santé Bio-innovation), Sainte Clotilde, Reunion Island, France. [8]Unité de Formation et de Recherche en Science et Technologie (UFR/ST), UniversitéNorbert ZONGO (UNZ), BP 376 Koudougou, Burkina Faso. [9]National Environment Agency, Singapore, Singapore. [10]SYSU Nuclear and Insect Biotechnology Co., Ltd, Dongguan, China. [11]Department of Microbiology and Molecular Genetics, Michigan State University, East Lansing, MI, USA. [12]ASTRE, CIRAD, F-34398 Montpellier, France. [13]ASTRE, Cirad, INRAE, Univ. Montpellier, Plateforme Technologique CYROI, Sainte-Clotilde, La Réunion, France. [14]These authors contributed equally: Dongjing Zhang, Hamidou Maiga, Yongjun Li, Mame Thierno Bakhoum, Gang Wang, Jérémy Bouyer. ✉e-mail: bouyer@cirad.fr

