## [Peer Review File · Nature Communications]

Mating harassment may boost the effectiveness of the sterile insect technique for *Aedes* mosquitoesReviewers' Comments:

Reviewer #1:

Remarks to the Author:

The study titled "Mating harassment may boost the effectiveness of the sterile insect technique for *Aedes* mosquitoes" by Zhang et al. presents relevant results about the effect of sexual harassment on the longevity of the *Aedes aegypti* and *Aedes albopictus* females, as well as the biting rates on humans. The results presented are of broad interest to the scientific community working on controlling vectors of human diseases, especially those using the Sterile Insect Technique (SIT) for mosquitoes control.

The document is well-written. The methods are explained in detail and in an understandable way. The conclusions are drawn based on the results presented, and their impact is well discussed.

The manuscript has some minor typos and redundant phrases. For example, line 166 should say "at" instead of and. In Supplementary Information, pg. 45, line 4, the word some is repeated twice.

I would like to point out that the term "native" (line 50) is wrongly used in this context. The SIT is not only used in the control of native insect species but also in the control of exotic ones. For example, SIT is used in different countries of the Americas to control the Mediterranean fruit fly, an exotic species. I suggest changing "native" to "wild" females.

Another imprecise term is "virgin" (line 50). Even though the main objective of the SIT is that sterile males mate with virgin wild females, it is also desirable that released males can mate with already mated females. For this reason, I also suggest removing the word "virgin". By using "virgin native" females, the authors wrongly constrain the SIT's extent.

Reviewer #2:

Remarks to the Author:

This paper explores whether inundative releases of sterile male *Aedes aegypti* and *Aedes albopictus* result in harassment of female mosquitoes, and suggests that this harassment can interfere with female feeding success, reducing their lifespan, host contact, and thereby pathogen transmission potential. The authors have explored this in a number of different lab and semi-field settings, as well as (for *Ae. albopictus*) in a field setting. This mechanism could explain why SIT and related control methods are potentially more effective than they would be based only on predictions regarding sterile mating. The main conclusions drawn from this work are presented as hypotheses regarding high male:female sex ratios reducing female feeding success, and that this may occur in a wider range of control methods that rely on distorting sex ratios. I think this is valuable and interesting, I do have the following comments regarding the study, which make it – to me – a bit challenging to clearly interpret the results as being due to harassment and reduced feeding success per se:

One of the concerns I have regarding the experimental design, is that although the sex ratio is changed among treatments, at the same time overall mosquito density is changed as well (e.g., page 27, line 125 – m:f numbers of 1980:20 and 20:20 were contrasted – this isn't completely clear everywhere, but it suggests that density was not kept at a constant level). From the methods, it is not clear why this decision was made, rather than varying sex ratio while keeping the overall density constant (presumably because it would be too cumbersome to perform the experiment otherwise). This potentially confounds the interpretation, as some of the results may then not have so much to do with male harassment per se, but could just be a consequence of density. This might explain for instance why in *Ae. albopictus* male mortality also increases with more biased sex ratios (although not in *Ae. aegypti*).

How the semi-field human landing catch was used to determine m:f ratios, or female feeding success is a bit unclear. Particularly, it is stated that a volunteer would kill mosquitoes as they land, but not described what this consists of or how this was performed. Particularly, I'd be concerned that with that many males flying around, it would become quite a difficult task to spot females as they're landing? In other words, how can you be sure females were less likely to approach or feed, rather than volunteers struggling to spot or kill them due to large number of males flying around their legs? At least a clearer description of the way these catches were performed would be helpful.

It's not clear how or whether the field trial can separate out effects related to feeding inhibition and harassment and an overall reduction in female numbers due to a reduction in population size. There is the more skewed sex ratio found in human landing catches compared to that found in BG traps, but it strikes me that it could be more helpful to compare differences in the proportion of eggs that hatched vs differences in abundance of eggs that hatched, since then you could potentially separate out differences in suppression due to sterility, versus suppression due to other effects on females. Was data on percent hatch rate in the different areas collected?

I do think the evidence for feeding inhibition from the lab is very interesting and raises the question whether and how this plays out in the field, and how that depends on the mating system of the species in question. For instance, could this suggest that SIT is actually potentially more effective against certain *Aedes* species than it is for *Anopheles* or *Culex*, due to the host-based mating system and possible disruption of feeding? What about species like *Ae. polynesiensis*, that appear to have swarms instead or in addition to host-based mating? Some discussion along these lines to put the current results and impact on SIT in the context of different mating habits of mosquitoes would be helpful.

The introduction is clear, and gives a good overview of the sterile insect technique and its potential for use to control *Aedes* spp. mosquitoes. There are a few citations that could potentially be added that have also touched on the issue of mating harassment in mosquitoes. For instance, Dao et al 2010 (*Journal of Medical Entomology*, 47(5), pp.769-777.) provides some evidence that also is suggestive of a cost of harassment, namely exposure to males, rather than consequences of mating, on female longevity. And Stone 2013 (*PLoS One* 8 (9), e76228) considers a number of aspects of male life history and mating behavior, including harassment of females, and explores their impact on SIT effectiveness using simple models.

Line 145 – "In nature, ..." This is speculative, and better kept for the discussion. One could also argue that in nature harassment might be much less important as females are more likely to be able to evade males, disperse to areas with fewer males, or possibly even shift their feeding behavior to hosts where males are less likely to aggregate?

In Fig 2, and the part of the results section that describes these results, please specify here (in addition to the methods) which species these results are based on. In general, the rationale for using both species for some, but only a single species, and not always the same, for other aspects of the study is not particularly clear. Replicating these studies among different locations and with different strains certainly is a strong point of the study, but at the same time, not everything is replicated everywhere, and being clear about the rationale behind these choices would strengthen the paper. For instance, why was the field trial performed with *Ae. albopictus*, rather than with *Ae. aegypti*?

Why are data not presented for the buffer zone? From fig 3 you had traps placed there as well, and this could tell you something about (possibly) whether females are avoiding / dispersing away from the release zone, or potentially whether your released males were having some impact in a surrounding area.

Line 249-254: This statement ("females that were exposed to males at a 1:3 or 99:1 ratio...did not show any increase in mortality") is unclear – from ext. data fig 5 it seems that both these treatments have a mortality cost associated with them, i.e., a lower survival than the virgin females of the same age. Doesn't that suggest the opposite, that it was the act of mating leading to this difference, rather than the act of harassment and possible energetic costs, which surely would have been much more intense at 99:1?

Reviewer #3:

Remarks to the Author:

Zhang et al study focused on the sterile insect technique (SIT) is a method of controlling insect pest populations that involves releasing large numbers of sterile males into the wild. The sterile males mate with wild females, but no offspring are produced, which helps to reduce the overall population of the pest species. However, recent research has shown that the release of sterile males in high ratios to females may also have unintended consequences.

Under laboratory conditions, they showed that female *Aedes* mosquitoes had reduced feeding success and shorter lifespans when exposed to male to female ratios above 50 to 1. Semi-field experiments also showed reduced blood uptake from artificial hosts and lower biting rates on humans. In a field trial conducted in China, the release of sterile males led to a reduction in female mosquito density and biting rates, but also resulted in increased female mortality due to mating harassment.

These findings suggest that while the SIT can effectively suppress mosquito populations and reduce disease transmission, it is important to carefully consider the ratio of sterile males to females in order to minimize unintended consequences such as increased female mortality and reduced feeding success.

Major comments:

1. The number of mosquitoes needed to be released continuously to achieve a certain level of reduction in the population would depend on several factors, such as the size of the target population, the release ratio of sterile males to wild females, and the frequency of releases. It would require a comprehensive analysis of the specific situation to determine the appropriate release strategy and the number of mosquitoes needed. How did these variables accounted in this study?
2. The manuscript does not provide information on the population of wild males or the size of natural swarms. The study mainly focuses on the impact of different ratios of sterile males to wild females on the target female population. How big was the swarm? Did they mark the swarming sites?
3. The manuscript does not provide information on the yearly biting rate and surveillance in the area before the release of sterile males. Could you please add more the details of the surveillance in that area before and after treatment?
4. Reaching the ratio of success would require careful planning and execution of the release strategy, taking into account factors such as the size of the area, the target population density, and the frequency of releases. It is possible that the SIT mosquitoes and natural mosquitoes could generate two different swarms, but this would depend on several factors, including the release strategy and the behaviour of the mosquitoes.
5. The graphs could be presented in a clearer manner with statistical significance accurately reported. This would help readers to better understand the level of accuracy of the data.
6. It is possible that the change in the location of the natural swarm may have contributed to the

sharp rise in the suppression observed in Figure 4a. However, further analysis would be required to confirm this.

7. The manuscript does not provide detailed information on the traps used or the reasons behind their selection. However, it is possible that alternative methods of capture could be explored to increase the number of mosquitoes captured and improve the statistical significance of the data.

8. The ethics are not clearly addressed. There is no information for ethics and protocol numbers, etc.

Minor comments:

The format of writing, whether US or British English, should be chosen and consistently applied throughout the manuscript. The entire manuscript should be revised accordingly to ensure adherence to the chosen format.

The lines that require revision for writing errors:

1. Page 2, lines 37-38
2. Page 30, line 200, "ad libitum" should be italicized
2. Page 45, line 4

REVIEWER COMMENTS

Reviewer #1 (Remarks to the Author):

The study titled “Mating harassment may boost the effectiveness of the sterile insect technique for *Aedes* mosquitoes” by Zhang et al. presents relevant results about the effect of sexual harassment on the longevity of the *Aedes aegypti* and *Aedes albopictus* females, as well as the biting rates on humans. The results presented are of broad interest to the scientific community working on controlling vectors of human diseases, especially those using the Sterile Insect Technique (SIT) for mosquitoes control.

The document is well-written. The methods are explained in detail and in an understandable way. The conclusions are drawn based on the results presented, and their impact is well discussed.

Many thanks for your positive comments.

The manuscript has some minor typos and redundant phrases. For example, line 166 should say “at” instead of and. In Supplementary Information, pg. 45, line 4, the word some is repeated twice.

OK, corrected

I would like to point out that the term “native” (line 50) is wrongly used in this context. The SIT is not only used in the control of native insect species but also in the control of exotic ones. For example, SIT is used in different countries of the Americas to control the Mediterranean fruit fly, an exotic species. I suggest changing “native” to “wild” females.

Agreed and corrected accordingly.

Another imprecise term is “virgin” (line 50). Even though the main objective of the SIT is that sterile males mate with virgin wild females, it is also desirable that released males can mate with already mated females. For this reason, I also suggest removing the word “virgin”.

By using “virgin native” females, the authors wrongly constrain the SIT’s extent.

Agreed and corrected accordingly.

Reviewer #2 (Remarks to the Author):

This paper explores whether inundative releases of sterile male *Aedes aegypti* and *Aedes albopictus* result in harassment of female mosquitoes, and suggests that this harassment can interfere with female feeding success, reducing their lifespan, host contact, and thereby pathogen transmission potential. The authors have explored this in a number of different lab and semi-field settings, as well as (for *Ae. albopictus*) in a field setting. This mechanism could explain why SIT and related control methods are potentially more effective than they would be based only on predictions regarding sterile mating. The main conclusions drawn from this work are presented as hypotheses regarding high male:female sex ratios reducing female feeding success, and that this may occur in a wider range of control methods that rely on distorting sex ratios. I think this is valuable and interesting, I do have the following comments regarding the study, which make it – to me – a bit challenging to clearly interpret the results as being due to harassment and reduced feeding success per se:

One of the concerns I have regarding the experimental design, is that although the sex ratio is changed among treatments, at the same time overall mosquito density is changed as well (e.g., page 27, line 125 – m:f numbers of 1980:20 and 20:20 were contrasted – this isn't completely clear everywhere, but it suggests that density was not kept at a constant level). From the methods, it is not clear why this decision was made, rather than varying sex ratio while keeping the overall density constant (presumably because it would be too cumbersome to perform the experiment otherwise). This potentially confounds the interpretation, as some of the results may then not have so much to do with male harassment per se, but could just be a consequence of density. This might explain for instance why in *Ae. albopictus* male mortality also increases with more biased sex ratios (although not in *Ae. aegypti*).

There is a misunderstanding here: in all lab trials aiming at measuring the impact of the ratio on the mortality, the overall density of mosquitoes was kept constant to avoid this confounding factor. We totally agree with the reviewer that otherwise, it would have impacted the mortality. It was already clearly stated in the Methods section (L46-50): "To study the sexual harassment of males on *Aedes* mosquito females without allowing potential density-dependent mortality, batches of 3,000 *Ae. aegypti* mosquitoes aged 0 to 1 day were placed in the Bugdorm cages (30 × 30 × 30 cm) at six male to female sex ratios (SR): SR= 3:7, SR = 1:3 (control, used in the colony maintained in mass-rearing conditions), SR = 10:1, SR = 23:2, SR = 49:1 and SR = 99:1." So there were always 3000 individuals per cage, and only the sex-ratio varied. To make it easier to understand by readers, we now have made it very clear in the main text, at the beginning of the section "Survival of mosquitoes caged at different sex ratios" by adding this sentence: "All experiments aiming to measure survival were done at a constant density of mosquitoes per cage, only varying the sex ratio, in order to control density-dependent mortality."

Now in the semi-field trial, we wanted to mimic what is happening in field conditions, where the release of large numbers of sterile males lead them to aggregate around human hosts, actually increasing the overall density of mosquitoes around hosts (swarms of males are visible and can even represent a nuisance). This time, we did not measure any mortality and we kept the vector (females) to host ratio constant to see how males can interfere with the vector-host contact measured by the engorgement rate for the artificial host, and the human landing catch (like in the field experiment) for the human host. We believe this was the right protocol to explore feeding interference? We added this sentence at the beginning of the corresponding section to guide the reader "In this experiment, we varied the male to female ratio while keeping the vector (female) to host ratio constant, in order to study the impact of mating harassment on host-vector contact."

How the semi-field human landing catch was used to determine m:f ratios, or female feeding success is a bit unclear. Particularly, it is stated that a volunteer would kill mosquitoes as they land, but not described what this consists of or how this was performed. Particularly, I'd be concerned that with that many males flying around, it would become quite a difficult task to spot females as they're landing? In other words, how can you be sure females were less likely to approach or feed, rather than volunteers struggling to spot or kill them due to large number of males flying around their legs? At least a clearer description of the way these catches were performed would be helpful.

We totally understand the concerns of the reviewers. It was indeed a challenging experiment and the volunteers were actually fully protected and killed all potentially successful females while not killing the males or unsuccessful females. We also checked at the end that there was no engorged females in the cages by collecting all of them. We now have included much more details in the Methods section as follows:

“We conducted a second experiment based on Human Landing Catch in China to assess whether male harassment can prevent blood feeding on humans in semi-field conditions. Wild type virgin *Ae. albopictus* (GUA strain) females were inseminated at 5-6 days old. They were starved for 24 hours before the experiment start. Irradiated HC males were virgin and 5-6 days old. Irradiated HC males were released into semi-field cages ($1.80 \times 1.80 \times 1.80$ m, containing two sugar water containers). GUA females were released 24 hours later into the semi-field cages. Male and female release numbers were 1980 versus 20 for the 99:1 ratio and 20 versus 20 for the 1:1 ratio. The mosquitoes were immobilized by placing them in a chilling room with a temperature of $8 \pm 2^\circ\text{C}$. Subsequently, we conducted a precise count of the required number of mosquitoes. Ten minutes after releasing the females, an adult volunteer wearing long-sleeved shirt, long pants, gloves, and mosquito-proof hat entered and sat on a chair in the middle of each cage. The collector exposed one of his legs from foot to knee and killed mosquitoes as soon as they landed on the exposed leg before they started feeding. Since there were female mosquitoes attempting to feed on the volunteers but failed due to mating harassment from males, only the females that successfully landed on exposed skin were classified as potential "successful bloodsuckers". To avoid the collection of male mosquitoes and "unsuccessful bloodsuckers", we opted to eliminate the landed females by swatting and killing them rather than using an aspirator. This approach was necessary due to the large number of male mosquitoes forming a swarm around the volunteer and the presence of female mosquitoes facing harassment while attempting to feed. This method prevented the collection of non-target mosquitoes. Mosquito collection was conducted for 15 min for each cage and ratio. All collected females were removed and counted. After 15 min of collection, remaining mosquitoes were collected with an aspirator and females individually checked for blood feeding or not. Three repeats were conducted with three different collectors managing one 99:1 and one 1:1 cage each. Collectors received appropriate information and gave their informed consent prior to participating in this study.”

We hope that the reviewer will be happy with these details.

It's not clear how or whether the field trial can separate out effects related to feeding inhibition and harassment and an overall reduction in female numbers due to a reduction in population size. There is the more skewed sex ratio found in human landing catches compared to that found in BG traps, but it strikes me that it could be more helpful to compare differences in the proportion of eggs that hatched vs differences in abundance of eggs that hatched, since then you could potentially separate out differences in suppression due to sterility, versus suppression due to other effects on females. Was data on percent hatch rate in the different areas collected?

We agree that the egg hatch rate can be potentially used to separate the difference. However, in this field trial we initially did not use the hatch rate because we encountered a problem with the methodology: the ovitraps were collected after 7 days of setting in the field and were left for another 7 days for incubation in the lab, but some of early laid eggs hatched in between and hatched larvae ate some of the eggs, which resulted in more larvae than eggs for some time/trap points (see raw data). We tried to conduct an analysis following your request, considering a hatching rate of 1 in all traps where the number of larvae exceeded the number of eggs (the number of larvae was considered equal to the number of eggs). We analysed the impact of the releases on the hatching rate using mixed binomial models with the time after the beginning of the releases, the treatment (release, buffer, control) and their first order interaction as fixed effects, and traps id and dates as random effect.

The graphs below present the dynamics of the hatch rate (and standard error) (the month is presented as a number and the releases started on 16th August):

We did not observe any significant effect of the treatment, meaning that all the observed suppression effect on both adults and eggs is probably due to mating harassment that can reduce female longevity and feeding success, as demonstrated in the other experiments, which will in turn result in reduced fecundity. We believe that the release area (1.17 ha) was too small and non isolated from the control area to make it possible to observe the effect of induced sterility, because fertile females could migrate from the control and buffer areas into the release area. However, such a small size with huge numbers of males was perfect to measure the effect of mating harassment, because sterile males do not move much. We made a similar observation during a MRR in Albania recently (Velo et al. 2022), where all males stayed <100m from the release site whereas we did not observe any spatial trend in induced sterility within 250m from the release point, suggesting stronger dispersal of egg-laying females than males.

We added this part to the results section:

“After the beginning of the releases, we did not observe any significant impact of the hatch rate of eggs between the release (0.43, min to max: 0.32-0.52) and the control (0.44, min to max: 0.36-0.52) areas ($z = 1.684$, $p = 0.092$), showing that induced sterility did not contribute to population suppression.”

We also acknowledged in the discussion that the hatching methodology is problematic and that we should not consider the absolute values. Removing all data points with more larvae than eggs also led to the same result:

Finally, we analysed all hatching data from 2022 and 2023 during this revision process and found similar results than in 2021 despite the correction of the hatching protocol, with no significant induced sterility in 2022, and a limited induced sterility of ~10% in 2023, unable to explain the observed suppression rates. Thanks for this remark that allowed to greatly improve the evidence of a strong impact of mating harassment on females.

I do think the evidence for feeding inhibition from the lab is very interesting and raises the question whether and how this plays out in the field, and how that depends on the mating system of the species in question. For instance, could this suggest that SIT is actually potentially more effective against certain *Aedes* species than it is for *Anopheles* or *Culex*, due to the host-based mating system and possible disruption of feeding? What about species like *Ae. polynesiensis*, that appear to have swarms instead or in addition to host-based mating? Some discussion along these lines to put the current results and impact on SIT in the context of different mating habits of mosquitoes would be helpful.

We totally agree with this remark and added this paragraph to the discussion:

“In both species studied here, feeding inhibition was demonstrated in the lab, together with an impact on female suppression in the field in the case of *Ae. albopictus*. However, such an impact will strongly depend on the mating system of the target species (Lees et al. 2014) and it would be important to study this phenomenon in *Anopheles* or *Culex* species, or even *Ae. polynesiensis* that may use swarms triggered by visual cues instead or in addition to host-based mating.”

The introduction is clear, and gives a good overview of the sterile insect technique and its potential for use to control *Aedes* spp. mosquitoes. There are a few citations that could potentially be added that have also touched on the issue of mating harassment in mosquitoes. For instance, Dao et al 2010 (Journal of Medical Entomology, 47(5), pp.769-777.) provides some evidence that also is suggestive of a cost of harassment, namely exposure to males, rather than consequences of mating, on female longevity. And Stone 2013 (PLoS One 8 (9), e76228) considers a number of aspects of male

life history and mating behavior, including harassment of females, and explores their impact on SIT effectiveness using simple models.

Many thanks for these references, which are indeed totally in line with our work! We added that of Dao et al. to the intro and that of Stone in the discussion.

Line 145 – “In nature, ...” This is speculative, and better kept for the discussion. One could also argue that in nature harassment might be much less important as females are more likely to be able to evade males, disperse to areas with fewer males, or possibly even shift their feeding behavior to hosts where males are less likely to aggregate?

OK, it was moved to the discussion section.

In Fig 2, and the part of the results section that describes these results, please specify here (in addition to the methods) which species these results are based on. In general, the rationale for using both species for some, but only a single species, and not always the same, for other aspects of the study is not particularly clear. Replicating these studies among different locations and with different strains certainly is a strong point of the study, but at the same time, not everything is replicated everywhere, and being clear about the rationale behind these choices would strengthen the paper. For instance, why was the field trial performed with *Ae. albopictus*, rather than with *Ae. aegypti*?

OK, we have now explained the rationale for species, strains and site selection in the relevant sections of the Methods. To be honest, the choice of the field trial was quite opportunistic to test our theory against at least one of the species because we took the opportunity of a preliminary SIT trial organized in China offering perfect settings to investigate the impact of male mating harassment in the absence of a strong induced sterility component, particularly a small and non-isolated release area. It is now specified in the Methods section.

Why are data not presented for the buffer zone? From fig 3 you had traps placed there as well, and this could tell you something about (possibly) whether females are avoiding / dispersing away from the release zone, or potentially whether your released males were having some impact in a surrounding area.

Thanks for this suggestion. We now included data from the buffer area in the results section of the manuscript. When we analysed the data on the number of females and the male/female ratio captured via BG and HLC in the buffer site, and compared to both release and control sites, we observed that the number of females captured showed no difference between the buffer and control sites, despite higher male/female ratios in the buffer than in the control area, but much lower than in the release site. This indicates that few sterile males dispersed to the buffer area and that the male to female ratio was insufficient to have impacts on the population density.

The following table summarizes the results for the period 3-6th Nov that we included in the paper:

Dates	No. of female capture via BG			Male/female ratio via BG		
	Release site	Buffer site	Control site	Release site	Buffer site	Control site
11.3-11.4	2.5 ± 1.3	3.40 ± 1.4	4.33 ± 0.9	12.5 ± 5.5	1.45 ± 0.7	0.44 ± 0.2
	No. of females	Male/female ratio via HLC				

	captured via HLC					
	Release site	Buffer site	Control site	Release site	Buffer site	Control site
11.3, 11.4 and 11.6	0.5 ± 0	3.33 ± 0.6	2.72 ± 0.2	101.3 ± 35.8	4.21 ± 1.1	0.96 ± 0.3

To better understand the threshold for feeding inhibition, we performed a new experiment to observe the effects of male/female ratio on the feeding rate as shown in the below table. We did not observe negative impact on the feeding rate when the male/female ratio was 10:1, as compared to 1:1. However, when the ratio of male/female reached 30:1, we observed a significant reduced feeding rate in a lab cage (30*30*30cm) with a mouse as a host. The average feeding rate was 16.7% in comparison to 56.7% in 1:1 ratio. This confirmed that a strong male to female ratio is necessary to observe feeding inhibition and we now included these results in the result section and fig. 2.

Line 249-254: This statement (“females that were exposed to males at a 1:3 or 99:1 ratio...did not show any increase in mortality”) is unclear – from ext. data fig 5 it seems that both these treatments have a mortality cost associated with them, i.e., a lower survival than the virgin females of the same age. Doesn’t that suggest the opposite, that it was the act of mating leading to this difference, rather than the act of harassment and possible energetic costs, which surely would have been much more intense at 99:1?

We agree with your interpretation and our sentence was unclear: we rephrased it.

Reviewer #3 (Remarks to the Author):

Zhang et al study focused on the sterile insect technique (SIT) is a method of controlling insect pest populations that involves releasing large numbers of sterile males into the wild. The sterile males mate with wild females, but no offspring are produced, which helps to reduce the overall population of the pest species. However, recent research has shown that the release of sterile males in high ratios to females may also have unintended consequences.

Under laboratory conditions, they showed that female Aedes mosquitoes had reduced feeding success and shorter lifespans when exposed to male to female ratios above 50 to 1. Semi-field experiments also showed reduced blood uptake from artificial hosts and lower biting rates on humans. In a field trial conducted in China, the release of sterile males led to a reduction in female mosquito density and biting rates, but also resulted in increased female mortality due to mating harassment.

These findings suggest that while the SIT can effectively suppress mosquito populations and reduce disease transmission, it is important to carefully consider the ratio of sterile males to females in order to minimize unintended consequences such as increased female mortality and reduced feeding success.

Thanks for this summary. There is however a misunderstanding in the last sentence: it is actually desirable to have increased female mortality and reduced feeding success because this will reduce disease transmission risk, even if it is an unintended effect.

Major comments:

1. The number of mosquitoes needed to be released continuously to achieve a certain level of reduction in the population would depend on several factors, such as the size of the target population, the release ratio of sterile males to wild females, and the frequency of releases. It would require a comprehensive analysis of the specific situation to determine the appropriate release strategy and the number of mosquitoes needed. How did these variables accounted in this study?

We totally agree with this remark and we even published a paper with such a recommendation (Bouyer et al. 2020). However, this study was a preliminary trial to set up procedures and measure the quality of the sterile males in order to prepare a suppression trial at a larger scale.

2. The manuscript does not provide information on the population of wild males or the size of natural swarms. The study mainly focuses on the impact of different ratios of sterile males to wild females on the target female population. How big was the swarm? Did they mark the swarming sites?

We indeed performed two mark-release-recapture (MRR) experiments to estimate wild mosquito population before release but we did not provide this information in the previous version of the manuscript. We now added one sentence in the maintext: “The density of the wild *Ae. albopictus* males was estimated to range from 6553 to 10076 males/ha and from 2875 to 5292 males/ha via two independent performed mark-release-recapture experiments performed just before the beginning of this trial (data not shown).”

Regarding to the swarm and swarm sites, we have to clarify that we did not observe “big swarms” observed in *Aedes albopictus*, but rather small diffused swarms upon hosts and visual markers. We thus did not mark swarming sites.

3. The manuscript does not provide information on the yearly biting rate and surveillance in the area before the release of sterile males. Could you please add more the details of the surveillance in that area before and after treatment?

Even though we did not continuously monitor the biting rate yearly, there were 4 independent human landing captures (HLCs) performed respectively in May, June, July and August before release (please see Extended Dada Fig. 6b). For the surveillance of mosquito population, we used ovitraps placed in the natural breeding sites from 8th March to 17th August before release (please see extended data Fig. 6a). Similar mosquito density in the release and un-release area based the number of the hatched eggs per ovitrap and the captured females (equal to biting index). The above-mentioned information has been shown in our Extended data Fig.6.

We now provided more detailed information on how we did the surveillance using the ovitraps. We added this paragraph in the Methods section:

“Briefly, the ovitrap constituted of transparent bottles with a black lid with three holes, allowing engorged females entering the trap for laying eggs. The ovitraps were cylindrical plastic containers of 70-75 mm diameter and 100 mm height. Before using, a piece of filter

paper (70 mm width and 45 mm height) was inserted along the ovitrap wall. A 50 mL bamboo leaf solution was added in the ovitrap to increase the trapping efficiency. Ovitrap were placed close to the natural breeding sites of *Ae. albopictus* for 7 days. The positive ovitrap were collected and incubated for another 7 days at room temperature before counting the number of eggs and the hatched larvae. Positive traps where eggs or larvae were observed were selected for further evaluation. The filter paper with eggs were removed from the ovitrap and the egg hatch rate was determined under a stereomicroscope. Boiling water was used to kill the remaining larvae and the number of larvae was counted. Unfortunately, we observed that some of early laid eggs hatched before initiating the hatching procedure and that larvae ate some of the floating eggs, which resulted in more larvae than eggs for some time-point data (see raw data). We thus considered a hatch rate of 1 for all these data points in the analysis.”

4. Reaching the ratio of success would require careful planning and execution of the release strategy, taking into account factors such as the size of the area, the target population density, and the frequency of releases. It is possible that the SIT mosquitoes and natural mosquitoes could generate two different swarms, but this would depend on several factors, including the release strategy and the behaviour of the mosquitoes.

As already presented in the manuscript based on both HLC and BG data, we indeed reached very high male to female ratios in our study (70.5:1 vs 16.6:1 respectively, Fig. 4d). As stated upon, there is no big swarms in *Aedes albopictus*, and the HLC collections do collect the males swarming upon human hosts where we saw that the mosquitoes were clearly mixed wild and sterile males using PCR.

5. The graphs could be presented in a clearer manner with statistical significance accurately reported. This would help readers to better understand the level of accuracy of the data.

OK, this was done as requested.

6. It is possible that the change in the location of the natural swarm may have contributed to the sharp rise in the suppression observed in Figure 4a. However, further analysis would be required to confirm this.

Again, there is not a single swarm in this species and we demonstrated that sterile and wild males were swarming together upon hosts.

7. The manuscript does not provide detailed information on the traps used or the reasons behind their selection. However, it is possible that alternative methods of capture could be explored to increase the number of mosquitoes captured and improve the statistical significance of the data. We used BG-Sentinel traps for collecting *Aedes albopictus* in this study since they are widely used for trapping *Aedes* mosquitoes (see studies Zheng X et al. *Nature*, 2019, 572(7767):56-61; Le Goff G et al. *Parasit Vectors*, 2019, 12(1):81; Le Goff G et al. *Parasit Vectors*, 2016, 9(1):514; Velo E et al. *Front Bioeng Biotechnol*, 2022, 10:833698.). To assess the biting index via human landing capture (HLC), fan trap is used to collect the surrounded mosquitoes which are attracted by the volunteers (as we did in a previous study, see (Zheng et al. 2019)). The description on how to use the fan trap is shown in the maintext in lines 241-243. Finally,

ovitrap were used to monitor the larval index. More detail information on the ovitrap has been added in the Method sections (see upon).

We fully agree with the Reviewer that different trapping methods will result in varied trapping efficiency. In our study, we used three important indicators via ovitrap, BG traps and HLC in both the release and control areas to ensure robust data. We believe that the three methods used in parallel were appropriate to give a good monitoring of the impact of the sterile male releases on the wild population.

8. The ethics are not clearly addressed. There is no information for ethics and protocol numbers, etc.

OK, this has been addressed.

Minor comments:

The format of writing, whether US or British English, should be chosen and consistently applied throughout the manuscript. The entire manuscript should be revised accordingly to ensure adherence to the chosen format.

OK, we applied English(UK) throughout our manuscript.

The lines that require revision for writing errors:

1. Page 2, lines 37-38
2. Page 30, line 200, "ad libitum" should be italicized
2. Page 45, line 4

OK, these typos have been corrected, thanks

References cited

- Bouyer, J., H. Yamada, R. Pereira, K. Bourtzis, and M. J. B. Vreysen. 2020.** Phased Conditional Approach for Mosquito Management using the Sterile Insect Technique. *Trends Parasitol.* 36: 325-336.
- Lees, R. S., B. G. J. Knols, R. Bellini, M. Q. Benedict, A. Bheecarry, H. Bossin, D. D. Chadee, J. P. Charlwood, R. K. Dabire, and L. Djogbenou. 2014.** Improving our knowledge of male mosquito biology in relation to genetic control programmes. *Acta Trop.* 132: S2-S11.
- Velo, E., F. Balestrino, P. Kadriaj, D. Carvalho, A. H. Dicko, R. Bellini, A. Puggioli, D. Petric, A. Michaelakis, F. Schaffner, D. Almenar, I. Pajovic, A. Beqirllari, M. Ali, G. Sino, E. Rogozi, V. Jani, A. Nikolla, T. Porja, T. Goga, E. Falcuta, M. Kavran, D. Pudar, O. Mikov, N. Ivanova-Aleksandrova, A. Cvetkovikj, M. M. Akiner, R. Mikovic, L. Tafaj, S. Bino, J. Bouyer, and W. Mamai. 2022.** A Mark-Release-Recapture study to estimate field performance of imported radio-sterilized male *Aedes albopictus* in Albania. *Frontiers in Bioengineering and Biotechnology* 10: 833698.
- Zheng, X., D. Zhang, Y. Li, C. Yang, Y. Wu, X. Liang, Z. Yan, L. Hu, Q. Sun, Y. Liang, J. Zhuang, X. Wang, Y. Wei, J. Zhu, W. Qian, A. G. Parker, J. R. L. Gilles, K. Bourtzis, J. Bouyer, M. Tang, J. Liu, Z. Hu, J.-T. Gong, X.-Y. Hong, Z. Zhang, L. Lin, Q. Liu, Z. Hu, Z. Wu, L. A. Baton, A. A. Hoffmann, and Z. Xi. 2019.** Incompatible and sterile insect techniques combined eliminate mosquitoes. *Nature* 572: 56-61.

Reviewers' Comments:

Reviewer #2:

Remarks to the Author:

The authors have thoroughly and clearly addressed all my concerns and I now recommend this paper be accepted.

Reviewer #3:

Remarks to the Author:

Thank you! I am happy with the responses.